# FILIP: Fine-grained Interactive Language-Image Pre-Training

**Lewei Yao** [1,2*]   **Runhui Huang**[3*]    **Lu Hou**[1*]    **Guansong Lu**[1]    **Minzhe Niu**[1]
**Hang Xu**[1†]   **Xiaodan Liang** [3†]    **Zhenguo Li**[1]    **Xin Jiang**[1]    **Chunjing Xu**[1]
[1]Huawei Noah's Ark Lab, [2]Hong Kong University of Science and Technology
[3]Sun Yat-sen University

## ABSTRACT

Unsupervised large-scale vision-language pre-training has shown promising advances on various downstream tasks. Existing methods often model the cross-modal interaction either via the similarity of the global feature of each modality which misses sufficient information, or finer-grained interactions using cross/self-attention upon visual and textual tokens. However, cross/self-attention suffers from inferior efficiency in both training and inference. In this paper, we introduce a large-scale Fine-grained Interactive Language-Image Pre-training (FILIP) to achieve finer-level alignment through a cross-modal late interaction mechanism, which uses a token-wise maximum similarity between visual and textual tokens to guide the contrastive objective. FILIP successfully leverages the finer-grained expressiveness between image patches and textual words by modifying only contrastive loss, while simultaneously gaining the ability to pre-compute image and text representations offline at inference, keeping both large-scale training and inference efficient. Furthermore, we construct a new large-scale image-text pair dataset called FILIP300M for pre-training. Experiments show that FILIP achieves state-of-the-art performance on multiple downstream vision-language tasks including zero-shot image classification and image-text retrieval. The visualization on word-patch alignment further shows that FILIP can learn meaningful fine-grained features with promising localization ability.

## 1 INTRODUCTION

Large-scale Vision-Language Pre-training (VLP) models like CLIP (Radford et al., 2021) and ALIGN (Jia et al., 2021) have recently demonstrated success across various downstream tasks. They learn visual and textual representations from millions of image-text pairs collected from the Internet and show superior zero-shot ability and robustness. The core technique of these models lies in the global contrastive alignment of the images and texts through a dual-stream model. Such architecture is inference-efficient for downstream tasks like retrieval because the encoders for the two modalities can be decoupled and the image or text representations can be pre-computed offline. However, CLIP and ALIGN model the cross-modal interaction via solely the similarity of the global feature of each modality, lacking the ability of capturing finer-level information like the relationship between visual objects and textual words. In this paper, we develop a simple yet efficient cross-modal finer-grained interaction mechanism for large-scale VLP.

To achieve finer-grained cross-modal interaction, previous methods mainly exploited two kinds of methods. (1) One line of work (Chen et al., 2020; Li et al., 2020b; Dong et al., 2021; Li et al., 2021b; Zhang et al., 2021; Zhan et al., 2021) uses a pre-trained object detector to extract region-of-interest (ROI) features from images, and then fuses it with the paired text through a VLP model. This design complicates the pre-training due to pre-computing and storing a large number of ROI features. In addition, the zero-shot ability of these approaches is usually limited by the predefined number of classes and their performance is also restricted by the quality of the detector. (2) Another line of work (Li et al., 2021a; Kim et al., 2021) enforces the token-wise or patch-wise representations from

---

*Equal contribution
†Corresponding authors: xu.hang@huawei.com, xdliang328@gmail.com

both modalities into the same space and models these finer-grained interactions via cross-attention (Li et al., 2021a) or self-attention (Kim et al., 2021). However, these methods are usually less efficient in terms of both training and inference. In particular, during training, cross-attention in (Li et al., 2021a) requires to be performed in an encoder-decoder structure, while the complexity of the self-attention (Kim et al., 2021) grows quadratically with the length of the prolonged concatenated sequences of both modalities. During inference, the data from both modalities are intertwined to compute the cross-attention or self-attention, and can not be pre-computed offline as dual-stream models like CLIP and ALIGN. This can be less efficient for downstream tasks like image/text retrieval and image classification.

In this paper, we propose a large-scale Fine-grained Interactive Language-Image Pre-training framework named FILIP to address these limitations. Inspired by Khattab & Zaharia (2020), we model the fine-grained semantic alignment through a novel cross-modal late interaction mechanism in the contrastive loss, instead of using cross or self-attention. Specifically, our fine-grained contrastive learning uses a token-wise maximum similarity between visual and textual tokens to guide the contrastive objective. In this way, FILIP successfully leverages the finer-grained expressiveness among image patches and textual words while simultaneously gaining the ability to pre-compute image and text representations offline. Unlike Khattab & Zaharia (2020), we discard the padded tokens and use average instead summation of token-wise maximum similarities when computing the image-text alignment, which enhances the cross-modal representation learning and stabilizes training. Furthermore, we construct a large-scale pre-training dataset named FILIP300M from the Internet. Data cleaning and image-text data augmentation are also explored and proved useful in this work.

Extensive experiments show that by effectively learning fine-grained representations, FILIP achieves state-of-the-art performance on multiple downstream tasks, including zero-shot image classification and image-text retrieval. For example, FILIP reaches 77.1% top-1 accuracy for zero-shot ImageNet classification, surpassing CLIP with less training data. Visualizations on word-patch alignment further show that FILIP learns meaningful finer-grained features with promising localization ability.

## 2 RELATED WORK

**Vision-Language Pre-training Models.** The pre-train-and-fine-tune scheme has achieved great success in the domains of natural language processing (Devlin et al., 2019; Brown et al., 2020) and computer vision (Dosovitskiy et al., 2020). It is then naturally extended to a joint cross-modal domain of Vision-and-Language Pre-training (VLP). The pre-training datasets of recent VLP models include publically available datasets like YFCC100M (Thomee et al., 2016) and CC12M (Changpinyo et al., 2021), as well as larger-scale datasets with more than 100M samples in CLIP (Radford et al., 2021) and ALIGN (Jia et al., 2021), which are shown to be even more powerful. The pre-training tasks of VLP models can be categorized into two categories: image-text contrastive learning task and Language Modeling (LM) based tasks: (i) CLIP (Radford et al., 2021), ALIGN (Jia et al., 2021) and UNIMO (Li et al., 2021b) make use of cross-modal contrastive learning which aligns the textual and visual information into a unified semantic space; (ii) VisualBERT (Li et al., 2019), UNITER (Chen et al., 2020), M6 (Lin et al., 2021), and DALL-E (Ramesh et al., 2021) employ LM-like objectives, including both masked LM (e.g., Masked Language/Region Modeling), and autoregressive LM (e.g., image captioning, text-grounded image generation). On the other hand, some methods rely on a pre-trained object detection model such as Faster-RCNN (Ren et al., 2015) to extract image regional features offline, which requires extra labeled bounding-box data and makes the approach less scalable. Recent efforts such as SOHO (Huang et al., 2021) and SimVLM (Wang et al., 2021) try to eliminate this burden via visual dictionary or PrefixLM (Raffel et al., 2020). In this paper, we directly learn fine-grained vision-language representations in an end-to-end and simpler manner while maintaining the benefit of inference efficiency.

**Multi-Modality Interaction Mechanism.** The core of vision-language pre-training models lies in modeling the interaction between the two modalities. There are mainly two types of cross-modal interaction architectures: Single-stream models like VisualBERT (Li et al., 2019) and ViLT (Kim et al., 2021) directly concatenate the patch-wise or regional visual features and textual embeddings and feed them to the transformer-based model. Dual-stream models such as ViLBERT (Lu et al., 2019) and CLIP (Radford et al., 2021) have separate encoders for different modalities. This allows flexible use of different models for different modalities, and efficient inference for downstream tasks like image-text retrieval, through the ability of decoupling the encoders and pre-compute image/text

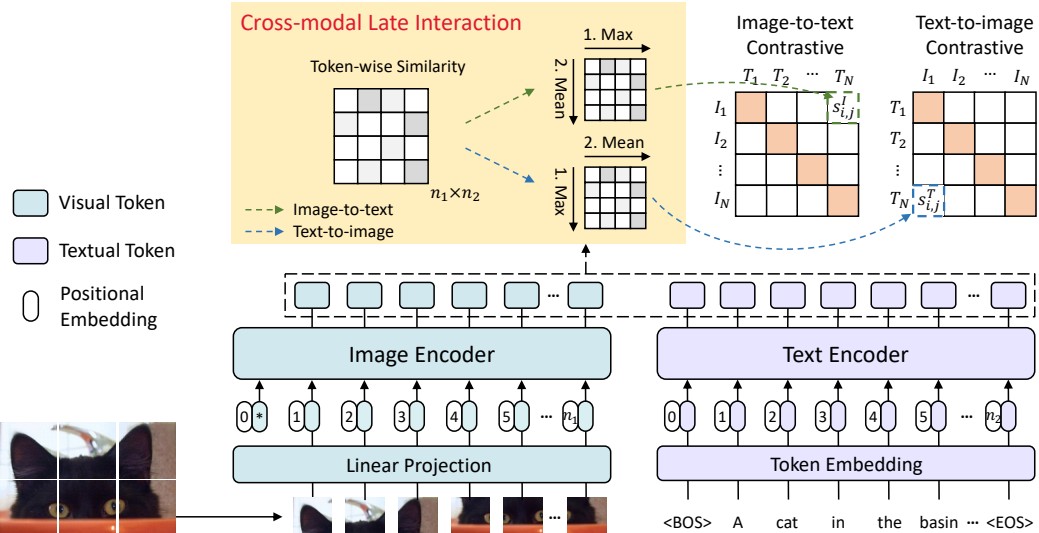

Figure 1: Overall architecture of FILIP, a dual-stream model with Transformer-based image and text encoders. On top of the image and text encoders, the representations of textual tokens and visual tokens are linearly projected to the multi-modal joint space. A novel fine-grained contrastive learning equipped with cross-modal late interaction is proposed, which uses a token-wise maximum similarity between visual and textual tokens.

features offline. SCAN (Lee et al., 2018) considers latent alignments between image regions and words. However, it is based on Triplet loss with a bottom-Up attention via a Faster-RCNN to extract object features while we try to directly learn to localize fine-grained object from patches. In this paper, while following the dual-stream approach for its flexible and efficient inference, we further propose a new multi-modal interaction mechanism to capture the fine-grained representations.

## 3 METHOD

In this paper, we propose a new cross-modal pre-training model that excels in fine-grained interaction between image encoder and text encoder for mining more detailed semantic alignment, named as FILIP, as shown in Figure 1. Particularly, FILIP is a dual-stream model with Transformer-based image and text encoders. For the visual modality, the image encoder is a Vision Transformer (Dosovitskiy et al., 2020) which takes the concatenation of an extra [CLS] token embedding and linearly projected image patches as input. For the textual modality, following Radford et al. (2021), we use the lower-cased byte pair encoding (BPE) (Sennrich et al., 2016b) with a vocabulary size of 49,408 to tokenize the text. Each text sequence starts with [BOS] token and ends with [EOS] token. After the word embedding layer, the token embeddings are fed into a modified decoder-only Transformer model as in (Radford et al., 2019). On top of the image and text encoders, the representations of textual tokens and visual tokens are linearly projected to the multi-modal common space, and are separately L2-normalized. Different from existing dual-stream models (e.g., CLIP and ALIGN) which models cross-modal interaction via only the global features of the entire image and text sequence, we introduce a novel fine-grained contrastive learning objective equipped with cross-modal late interaction which takes into account the fine-grained interaction between image patches and textual tokens, detailed in Section 3.1.

### 3.1 FINE-GRAINED CONTRASTIVE LEARNING

Contrastive representation learning has recently been found to learn better representations than its predictive counterpart in both visual (Tian et al., 2020) and vision-language cross-modal pre-training (Radford et al., 2021). Under a general formulation of cross-modal contrastive learning (Radford et al., 2021), we want to learn encoders $f_\theta$ for image data $\mathcal{I}$ and $g_\phi$ for text data $\mathcal{T}$ such that, given an image $\boldsymbol{x}^I \in \mathcal{I}$, and a text $\boldsymbol{x}^T \in \mathcal{T}$, the encoded representations $f_\theta(\boldsymbol{x}^I)$ and $g_\phi(\boldsymbol{x}^T)$ are close if they are related and far apart if not, under a distance metric. In each training batch, we sample $b$ image-text pairs $\{\boldsymbol{x}_k^I, \boldsymbol{x}_k^T\}_{k=1}^b$, For image $\boldsymbol{x}_k^I$ in image-text pair $\{\boldsymbol{x}_k^I, \boldsymbol{x}_k^T\}$, $\boldsymbol{x}_k^T$ is its positive, while

the other texts will be used as in-batch negatives. The image-to-text contrastive loss $\mathcal{L}_k^I$ for $\boldsymbol{x}_k^I$ can then be formulated as

$$\mathcal{L}_k^I(\boldsymbol{x}_k^I, \{\boldsymbol{x}_j^T\}_{j=1}^b) = -\frac{1}{b} \log \frac{exp(s_{k,k}^I)}{\sum_j exp(s_{k,j}^I)},$$

where $s_{k,j}^I$ denotes the similarity of the $k$-th image to the $j$-th text. Similarly, the text-to-image contrastive loss for $\boldsymbol{x}_k^T$ is

$$\mathcal{L}_k^T(\boldsymbol{x}_k^T, \{\boldsymbol{x}_j^I\}_{j=1}^b) = -\frac{1}{b} \log \frac{exp(s_{k,k}^T)}{\sum_j exp(s_{j,k}^T)}.$$

The total loss of this mini-batch can be represented by

$$\mathcal{L} = \frac{1}{2} \sum\nolimits_{k=1}^b (\mathcal{L}_k^I + \mathcal{L}_k^T). \tag{1}$$

### 3.1.1 Cross-modal Late Interaction

From the contrastive loss (1), the cross-modal interaction is reflected in how we compute the similarities $s_{i,j}^I$ and $s_{i,j}^T$ for the $i$-th image and $j$-th text. Previous methods like CLIP (Radford et al., 2021) and ALIGN (Jia et al., 2021) simply encode each image or text separately to a global feature i.e., $f_\theta(\boldsymbol{x}_i^I) \in \mathbb{R}^d$ and $g_\phi(\boldsymbol{x}_j^T) \in \mathbb{R}^d$, and compute these two similarities as

$$s_{i,j}^I = s_{i,j}^T = f_\theta(\boldsymbol{x}_i^I)^\top g_\phi(\boldsymbol{x}_j^T), \tag{2}$$

neglecting finer-grained interactions (e.g., word-patch alignment) between the two modalities. To alleviate this problem, while simultaneously maintain the training and inference efficiency of dual-stream models, we apply a cross-modal late interaction inspired by Khattab & Zaharia (2020) to model the token-wise cross-modal interaction.

Specifically, denote $n_1$ and $n_2$ as the number of (non-padded) tokens of the $i$-th image and $j$-th text, respectively, and the corresponding encoded features are $f_\theta(\boldsymbol{x}_i^I) \in \mathbb{R}^{n_1 \times d}$ and $g_\phi(\boldsymbol{x}_j^T) \in \mathbb{R}^{n_2 \times d}$. For the $k$-th visual token, we compute its similarities with all textual tokens of $\boldsymbol{x}_j^T$, and use the largest one

$$\max_{0 \leq r < n_2} [f_\theta(\boldsymbol{x}_i^I)]_k^\top [g_\phi(\boldsymbol{x}_j^T)]_r \tag{3}$$

as its token-wise maximum similarity with $\boldsymbol{x}_j^T$. We then use the average token-wise maximum similarity of all non-padded tokens in the image (resp. text) as the similarity of an image to a text (resp. a text to an image). The similarity of the $i$-th image to the $j$-th text can thus be formulated as:

$$s_{i,j}^I(\boldsymbol{x}_i^I, \boldsymbol{x}_j^T) = \frac{1}{n_1} \sum_{k=1}^{n_1} [f_\theta(\boldsymbol{x}_i^I)]_k^\top [g_\phi(\boldsymbol{x}_j^T)]_{m_k^I}, \tag{4}$$

where $m_k^I = \arg\max_{0 \leq r < n_2} [f_\theta(\boldsymbol{x}_i^I)]_k^\top [g_\phi(\boldsymbol{x}_j^T)]_r$. Similarly, the similarity of the $j$-th text to the $i$-th image is

$$s_{i,j}^T(\boldsymbol{x}_i^I, \boldsymbol{x}_j^T) = \frac{1}{n_2} \sum_{k=1}^{n_2} [f_\theta(\boldsymbol{x}_i^I)]_{m_k^T}^\top [g_\phi(\boldsymbol{x}_j^T)]_k, \tag{5}$$

where $m_k^T = \arg\max_{0 \leq r < n_1} [f_\theta(\boldsymbol{x}_i^I)]_r^\top [g_\phi(\boldsymbol{x}_j^T)]_k$. Note that $s_{i,j}^I(\boldsymbol{x}_i^I, \boldsymbol{x}_j^T)$ in Equation (4) does not necessarily equal $s_{i,j}^T(\boldsymbol{x}_i^I, \boldsymbol{x}_j^T)$ in Equation (5).

**Remark 1** *Intuitively, the token-wise maximum similarity in Equation (3) means that for each image patch, we find its most similar textual token. Similarly, for each textual token, we also find its closest image patch. By applying this to the similarity calculation in (4) and (5) for contrastive loss (1), the dual-stream model learns fine-grained alignment between image patches and textual tokens.*

The original late interaction mechanism in (Khattab & Zaharia, 2020) computes the relevance score of a document to a query *padded with mask tokens*, as a *sum* of token-wise maximum similarities, and is optimized via a *pairwise* softmax cross-entropy loss. Though inspired from Khattab & Zaharia (2020), our proposed cross-modal late interaction differs in several aspects. Firstly, we

exclude the padded textual tokens when computing the similarity, as they harm the performance. We speculate that this is because these padded tokens also learn textual representations and will mislead the model to align image patches to these meaningless padded tokens rather than meaningful non-padded words. Secondly, when computing similarities (4) and (5), we use the average of the token-wise maximum similarities instead of summation in (Khattab & Zaharia, 2020). This is because the number of non-padded tokens varies from text to text, and this summation over all non-padded tokens can have quite different magnitudes, leading to less stabilized training and worse final performance. These two modifications are crucial to not only the downstream tasks' performance, but also the quality of the word-patch alignment. A more detailed discussion can be found in Appendix A.7. Thirdly, we optimize the late interaction mechanism via a contrastive loss (1) which is found powerful vision-language pre-training (Radford et al., 2021) instead of the original pairwise loss in (Khattab & Zaharia, 2020).

**Training Efficiency.** Though the cross-modal late interaction is able to capture finer-grained features compared with the original loss, it relies on the token-wise representations of both modalities, and can be inefficient in terms of communication, memory and computation, especially when the batch size is large. To alleviate this problem, we utilize several methods. Firstly, we reduce the embedding size to 256. Besides, we reduce the precision of the last-layer features of both modalities from fp32 to fp16 before node communication in a distributed learning setting, and perform the multiplication in Equations (4) and (5) under the reduced precision. In addition, since the complexity of similarity calculation scales with the sequence length of textual tokens and image patches, for each image (resp. text), we select the 25% tokens with the highest token-wise maximum similarity score (Equation (3)) among all texts (resp. images) in the same local worker before node communication, based on the intuition that each sample can be represented by a few of the most representative tokens. Effects of these modifications are studied in Section 4.4.

### 3.1.2 PROMPT ENSEMBLE AND TEMPLATES

Due to the problem of polysemy and inconsistency with the pre-training process, following Radford et al. (2021), we also use prompt templates to augment the original label for some downstream tasks. For visualizations, for simplicity, we use only one prompt template across the paper, i.e. "a photo of a {label}." as Radford et al. (2021). For other experiments, we report results using prompt ensemble following Radford et al. (2021). When multiple prompts are allowed, the token-wise representations of different prompt templates for the same class label are different, and can not be summed together to form a mean textual representation as in (Radford et al., 2021). Thus, instead of ensembling different prompt templates by their mean textual representation, we ensemble them by their mean token-wise similarity. Specifically, suppose there are $C$ prompt templates, each label is augmented to $C$ different texts $\boldsymbol{x}_1^T, \boldsymbol{x}_2^T, \cdots, \boldsymbol{x}_C^T$. The similarity between an image $\boldsymbol{x}^I$ and this label is computed as $\frac{1}{C} \sum_{c=1}^{C} s_{\cdot,\cdot}^I(\boldsymbol{x}^I, \boldsymbol{x}_c^T)$, where $s_{\cdot,\cdot}^I$ is defined in Equation (4).

We use a unified rule-based method inspired by Radford et al. (2018) to construct prompt templates for image classification tasks. Specifically, each template consists of four components:

$$[\text{prefix}] \{\text{label}\}, [\text{category description}]. [\text{suffix}]. \tag{6}$$

Here, the "[prefix]" is an in-context description like "a photo of a" similar as Radford et al. (2021); "label" is a class label of the dataset; "[category description]" describes the category which is found helpful for some fine-grained image classification datasets (Radford et al., 2021), e.g., " a type of pet" for dataset Oxford-IIIT Pets. An interesting finding is that, adding a suffix that includes the reference word "it" (e.g., "I like it.") at the end of the prompt empirically improves the zero-shot classification performance of the proposed model. We speculate this is because the reference word "it" strengthens the fine-grained cross-modal alignment, as it can also be aligned to image patches of the target object. Detailed prompt templates for different datasets can be found in Appendix A.5.

### 3.2 IMAGE AND TEXT AUGMENTATION

To obtain better generalization and data-efficiency of the model, we perform data augmentation on both images and texts during the pre-training phase to construct more image-text pairs. We apply AutoAugment (Krizhevsky et al., 2012; Sato et al., 2015; Cubuk et al., 2019; Hoffer et al., 2020) for image augmentation, following the SOTA vision recognition methods (Touvron et al., 2021; Xie et al., 2020b). To ensure the augmented texts are semantically similar as the original one, for text

Table 1: Top-1 accuracy(%) of zero-shot image classification on 12 datasets. Our FILIP can boost 3∼5% accuracy on average.

| | CIFAR10 | CIFAR100 | Caltech101 | StanfordCars | Flowers102 | Food101 | SUN397 | DTD | Aircrafts | OxfordPets | EuroSAT | ImageNet | Average |
|---|---|---|---|---|---|---|---|---|---|---|---|---|---|
| CLIP-ViT-B/32 | 91.3 | 65.1 | 87.9 | 59.4 | 66.7 | 84.4 | 63.2 | 44.5 | 21.2 | 87.0 | 49.4 | 63.2 | 65.3 |
| FILIP$_{base}$-ViT-B/32 | 86.9 | 65.5 | 91.9 | 55.4 | 85.3 | 82.8 | 69.1 | 49.3 | 57.2 | 88.1 | 49.9 | 68.8 | **70.9**$^{+5.6}$ |
| CLIP-ViT-L/14 | 96.2 | 77.9 | 92.6 | 77.3 | 78.7 | 92.9 | 67.7 | 55.3 | 36.1 | 93.5 | 59.9 | 75.3 | 75.3 |
| FILIP$_{large}$-ViT-L/14 | 95.7 | 75.3 | 93.0 | 70.8 | 90.1 | 92.2 | 73.1 | 60.7 | 60.2 | 92 | 59.2 | 77.1 | **78.3**$^{+3.0}$ |

augmentation, we rewrite the original text using back-translation (Xie et al., 2020a; Sennrich et al., 2016a). Specifically, the texts are first translated to the target language and then translated back to the source language. We choose German and Russian as the target language and get extra two texts for each image-text pair. When constructing a batch of image-text pairs during the pre-training, the text of each image-text pair is randomly sampled from the three candidate texts, i.e., the original text and two back-translated texts.

## 3.3 PRE-TRAINING DATASET

A sufficiently large image-text dataset is a prerequisite for vision-language pre-training. Recent CLIP (Radford et al., 2021) and ALIGN (Jia et al., 2021) construct datasets with 400M and 1800M image-text pairs, respectively. In this work, we also collect a large-scale dataset called FILIP300M from the Internet, which consists of 300M image-text pairs and covers board vision and language concepts. For image-based filtering, we remove the images whose shorter dimension is smaller than 200 pixels and the aspect ratio is larger than 3. For text-based filtering, we keep only English texts, and exclude the meaningless ones, e.g., img_0.jpg. We also discard image-text pairs whose texts are repeated for over 10 times. Besides, we also use 3 public datasets, including Conceptual Captions 3M (CC3M) (Sharma et al., 2018), Conceptual 12M (CC12M) (Changpinyo et al., 2021) and Yahoo Flickr Creative Commons 100M (YFCC100M) (Thomee et al., 2016). We apply the same filtering rules on YFCC100M. Finally, we use about 340M image-text pairs for pre-training. Despite using a smaller training dataset than CLIP and ALIGN, our models still outperform them in most down-steam tasks (see Section 4).

## 4 EXPERIMENTS

### 4.1 EXPERIMENTAL SETUP

**Model Architectures.** We train two models from scratch, i.e., FILIP$_{base}$ and FILIP$_{large}$. The model architectures follow CLIP (Radford et al., 2021), i.e., the image encoder is ViT-B/32 for FILIP$_{base}$ and ViT-L/14 for FILIP$_{large}$. More details can be found in Appendix A.3.

**Pre-training Details.** To save memory and scale up the batch size, automatic mixed-precision (Micikevicius et al., 2018) and gradient checkpoint (Griewank & Walther, 2000; Chen et al., 2016) are used The input images are resized to $224 \times 224$ resolution during pre-training and the maximum length of the text is limited to 77 tokens following Radford et al. (2021). The training is mainly conducted on Nvidia V100 GPUs and Ascend Cards. FILIP$_{base}$ is trained on 128 cards about 9 days and FILIP$_{large}$ takes about 24 days to train on 192 cards. Unless otherwise specified, we use FILIP$_{large}$ to compare with other methods and FILIP$_{base}$ for ablation. We train both models using the LAMB optimizer (You et al., 2020) and cosine learning rate schedule (Loshchilov & Hutter, 2016) with a linear warmup. Weight decay regularization is applied to all parameters except bias, layer normalization, token embedding, positional embedding and temperature in contrastive loss. Detailed values of hyperparameters for different datasets and models can be found in Appendix A.3.

### 4.2 IMAGE CLASSIFICATION

In this section, we compare our FILIP with CLIP (Radford et al., 2021) on 12 downstream image classification datasets.

Table 2: Results of zero-shot image-text retrieval on Flickr30K and MSCOCO datasets. The last two rows (marked with *) report the zero-shot results on Flickr30K dataset of model fine-tuned on MSCOCO dataset, following the setting of ALBEF (Li et al., 2021a).

| | Flickr30K | | | | | | MSCOCO | | | | | |
| | image-to-text | | | text-to-image | | | image-to-text | | | text-to-image | | |
| | R@1 | R@5 | R@10 | R@1 | R@5 | R@10 | R@1 | R@5 | R@10 | R@1 | R@5 | R@10 |
|---|---|---|---|---|---|---|---|---|---|---|---|---|
| Unicoder-VL | 64.3 | 85.8 | 92.3 | 48.4 | 76.0 | 85.2 | – | – | – | – | – | – |
| ImageBERT | 70.7 | 90.2 | 94.0 | 54.3 | 79.6 | 87.5 | 44.0 | 71.2 | 80.4 | 32.3 | 59.0 | 70.2 |
| UNITER | 83.6 | 95.7 | 97.7 | 68.7 | 89.2 | 93.9 | – | – | – | – | – | – |
| CLIP | 88.0 | 98.7 | 99.4 | 68.7 | 90.6 | 95.2 | 58.4 | 81.5 | 88.1 | 37.8 | 62.4 | 72.2 |
| ALIGN | 88.6 | 98.7 | 99.7 | **75.7** | **93.8** | **96.8** | 58.6 | 83.0 | 89.7 | 45.6 | 69.8 | 78.6 |
| **FILIP** | **89.8** | **99.2** | **99.8** | 75.0 | 93.4 | 96.3 | **61.3** | **84.3** | **90.4** | **45.9** | **70.6** | **79.3** |
| ALBEF* | 94.1 | 99.5 | 99.7 | 82.8 | 96.3 | 98.1 | – | – | – | – | – | – |
| **FILIP*** | **95.4** | **99.8** | **100.0** | **84.7** | **97.0** | **98.7** | – | – | – | – | – | – |

**Zero-shot Classification.** As in Section 3.1.2, we apply a set of prompts (Appendix A.5) for each dataset and ensemble them to get the final results. Table 1 shows the results on 12 datasets. Despite using less training data (340M vs. 400M), both FILIP$_{base}$ and FILIP$_{large}$ considerably outperform their CLIP counterparts in terms of average top-1 accuracy over 12 datasets, i.e., achieving absolute improvements of 5.6% and 3.0%, respectively. In particular, our FILIP surpasses CLIP on ImageNet, the largest dataset among 12 datasets. FILIP also achieves substantial performance gains on some domain-specific datasets like Aircrafts. We speculate this is because, unlike CLIP which aggregates the information of the whole image into the [CLS] token, our proposed FILIP focuses more on the target object by directly aligning the image patches of the target object with the textual tokens corresponding to the class label (visualizations of word-patch alignment are in Section 4.5).

**Linear Probe.** Table 14 in Appendix A.6 shows the linear probe results, and FILIP again outperforms CLIP by 1.2~1.8% points on average. More details can be found in Appendix A.6.

## 4.3 IMAGE-TEXT RETRIEVAL

Image-text retrieval consists of two sub-tasks: image-to-text retrieval and text-to-image retrieval. We evaluate our FILIP model on two retrieval benchmark datasets: Flickr30K (Plummer et al., 2015) and MSCOCO (Lin et al., 2014), under both zero-shot and fine-tuned settings. More details of experimental setting can be found in Appendix A.3.

Tables 2 and 3 show the results of zero-shot and fine-tuned image-text retrieval, respectively. We compare our FILIP model against methods with complex attention layers including Unicoder-VL (Li et al., 2020a), ImageBERT (Qi et al., 2020), UNITER (Chen et al., 2020), VILLA (Gan et al., 2020), ERNIE-ViL (Yu et al., 2021), Oscar (Li et al., 2020b), VinVL (Zhang et al., 2021), ALBEF (Li et al., 2021a), and methods trained on larger-scale image-text datasets including CLIP (Radford et al., 2021) and ALIGN (Jia et al., 2021). As we can see, FILIP achieves state-of-the-art performances under all metrics on both Flickr30K and MSCOCO datasets, except for zero-shot text-to-image retrieval on Flickr30K, where FILIP achieves competitive performance with SOTA. For zero-shot image-to-text retrieval on MSCOCO dataset, the absolute R@1 of our proposed FILIP is 2.7% higher than ALIGN, which is trained on a much larger dataset.

## 4.4 ABLATION STUDY

**Effectiveness of Each Component.** We study the effectiveness of each component in FILIP, i.e., image/text augmentations and cross-modal late interaction. Experiments are conducted on FILIP$_{base}$, with a filtered subset of YFCC100M as the training dataset (as described in Section 3.3), on both zero-shot retrieval and classification tasks. We measure models' performance on MSCOCO zero-shot image-text retrieval and ImageNet zero-shot classification, which are two effective indicators for the quality of the learned vision-language representations.

Table 4 reports the results. As can be seen, all three components are beneficial for both tasks. Despite the simple design, cross-modal late interaction brings significant performance improvements over the baseline (the vanilla CLIP ViT-B/32), with an absolute R@1 gain of 5.5% (resp. 3.8%) for image-to-text (resp. text-to-image) retrieval on MSCOCO and an absolute top-1 accuracy gain

Table 3: Results of fine-tuned image-text retrieval on Flickr30K and MSCOCO datasets.

| | Flickr30K | | | | | | MSCOCO | | | | | |
| | image-to-text | | | text-to-image | | | image-to-text | | | text-to-image | | |
| | R@1 | R@5 | R@10 | R@1 | R@5 | R@10 | R@1 | R@5 | R@10 | R@1 | R@5 | R@10 |
|---|---|---|---|---|---|---|---|---|---|---|---|---|
| Unicoder-VL | 86.2 | 96.3 | 99.0 | 71.5 | 90.9 | 94.9 | 62.3 | 87.1 | 92.8 | 48.4 | 76.7 | 85.9 |
| ImageBERT | 87.0 | 97.6 | 99.2 | 73.1 | 92.6 | 96.0 | 66.4 | 89.8 | 94.4 | 50.5 | 78.7 | 87.1 |
| UNITER | 87.3 | 98.0 | 99.2 | 75.6 | 94.1 | 96.8 | 65.7 | 88.6 | 93.8 | 52.9 | 79.9 | 88.0 |
| VILLA | 87.9 | 97.5 | 98.8 | 76.3 | 94.2 | 96.8 | – | – | – | – | – | – |
| ERNIE-ViL | 88.1 | 98.0 | 99.2 | 76.7 | 93.6 | 96.4 | – | – | – | – | – | – |
| Oscar | – | – | – | – | – | – | 73.5 | 92.2 | 96.0 | 57.5 | 82.8 | 89.8 |
| VinVL | – | – | – | – | – | – | 75.4 | 92.9 | 96.2 | 58.8 | 83.5 | 90.3 |
| ALIGN | 95.3 | 99.8 | 100.0 | 84.9 | 97.4 | 98.6 | 77.0 | 93.5 | 96.9 | 59.9 | 83.3 | 89.8 |
| ALBEF | 95.9 | 99.8 | **100.0** | 85.6 | 97.5 | 98.9 | 77.6 | 94.3 | 97.2 | 60.7 | **84.3** | 90.5 |
| Our FILIP | **96.6** | **100.0** | **100.0** | **87.1** | **97.7** | **99.1** | **78.9** | **94.4** | **97.4** | **61.2** | **84.3** | **90.6** |

Table 4: Ablation study of different components on pre-training subset of YFCC100M. I2T and T2I are abbreviations for image-to-text and text-to-image retrieval, respectively. "ZS" means zero-shot performance. Underlined numbers have the highest improvements for the corresponding metrics.

| Model | MSCOCO | | | | ImageNet |
| | I2T R@1 | I2T R@5 | T2I R@1 | T2I R@5 | ZS Top1 |
|---|---|---|---|---|---|
| Baseline (ViT-B/32) | 25.0 | 49.5 | 14.7 | 34.7 | 30.4 |
| w/ image augmentation | 26.1 | 51.8 | 16.5 | 37.5 | 32.5 |
| w/ back translation | 29.2 | 55.0 | 17.9 | 39.8 | 33.9 |
| w/ cross-modal late interaction | 30.5 | 55.3 | 18.5 | 40.0 | 34.3 |
| Our FILIP$_{base}$ | **33.4** | **60.1** | **23.0** | **46.2** | **37.8** |

Table 5: Efficiency study of the cross-modal late interaction. "orig" and "late" stand for the contrastive loss based on the original cosine similarity in CLIP and our proposed cross-modal late interaction, respectively. "ZS" means zero-shot performance. We report results for ViT-B/32 trained on filtered YFCC100M with 8 V100 GPUs, with a batch size of 512 per GPU. Training time and memory consumption are tested using the same gradient checkpoint configuration. * denotes our final setting used in other experiments.

| Loss | Embed dim | Embed precision | Token % | Training time (sec/iter) | Memory (MB) | ImageNet ZS Top1 |
|---|---|---|---|---|---|---|
| orig (baseline) | 512 | fp32 | - | 1.31 | 14300 | 30.4 |
| late | 512 | fp32 | 100% | 2.85 | 26000 | 34.6 |
| late | 512 | fp16 | 100% | 2.67 | 23468 | 34.5 |
| late | 256 | fp16 | 100% | 2.31 | 22382 | **35.2** |
| late | 256 | fp16 | 50% | 1.61 | 16336 | 34.5 |
| late* | 256 | fp16 | 25% | 1.39 | 16100 | 34.3 |

of 3.9% for zero-shot classification on ImageNet. Further improvements are observed when all components are combined together.

**Efficiency Study of Cross-modal Late Interaction.** Since the late interaction mechanism in Section 3.1.1 requires to calculate the similarity between all visual and textual tokens, its efficiency can be a problem when employed in large-scale distributed training. As described in Section 3.1.1, we make several attempts to address the issue. Table 5 shows the efficiency improvement on zero-shot classification on ImageNet when these attempts are applied. As can be seen, these attempts improve the efficiency of late interaction without accuracy drop. Combining all three attempts achieves only slightly slower training and larger memory consumption than the original loss in CLIP.

## 4.5  VISUALIZATION OF FINE-GRAINED ALIGNMENT

In this section, we visualize FILIP's capability of capturing fine-grained cross-modal correspondence using the method of word-patch alignment. To make a fair comparison, we use our FILIP$_{base}$ trained on YFCC100M and CLIP's ViT-B/32, which are of the same size, for visualization. Each image is patchified to $7 \times 7$ image patches. More visualization results can be found in Appendix A.4.

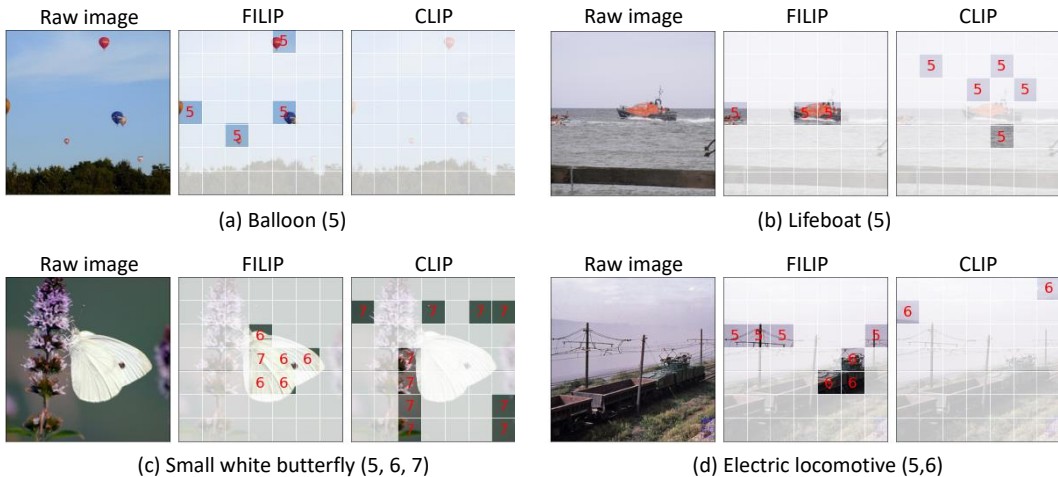

Figure 2: Visualizations of word-patch alignment for 4 classes of the ImageNet dataset and "a photo of a {label}." is the prompt. Numbers in the parentheses after the class label indicate the location indices of the class label in the tokenized textual sequence. The correct predictions are highlighted by opaque patches with the class label indices in red.

**Visualization Method.** The word-patch alignment is performed based on the token-wise similarity between the image patches and textual tokens. Specifically, for the $k$-th image patch, the location index of textual token with the largest similarity with it ($m_k^I$ in Equation (4)) is considered as its predicted label, and is placed at the center of it. Take class "balloon" as an example. There are 8 tokens in the tokenized textual sequence "[BOS] a photo of a balloon. [EOS]", and the location index of the class label "balloon" is "5". Note that one class label may be tokenized to more than one token. Location indices of textual tokens corresponding to the class label are highlighted in red, while the others are marked in white. A desired model that learns fine-grained representations would predict image patches of the target object to red indices.

**Observations.** Figure 2 shows the word-patch alignment results for FILIP and CLIP on 4 classes from the ImageNet dataset. As can be seen, FILIP exhibits the finer-grained understanding of an image in the following aspects. (i) A single object: From the visualization of class "small white butterfly", the image patches covering the object are all classified correctly; (ii) Same object in different shapes: From the visualizations of class "balloon" and "lifeboat", image patches corresponding to all target objects with different shapes and locations are correctly classified; (iii) Key Components of an object: For class "electric locomotive", there are two key components crucial to correctly classifying the image, i.e., "electric" and "locomotive", whose corresponding textual token indices are "5" and "6", respectively. As can be seen, image patches matching these two key components are respectively correctly classified. On the other hand, CLIP can not correctly align image patches with corresponding textual tokens. Compared with Kim et al. (2021) which uses an extra optimal transport to align the textual word and image patch distributions, the word-patch alignment can be simply automatically learned by our method.

## 5   CONCLUSION AND FUTURE WORK

This paper introduces FILIP, a simple yet generic framework towards fine-grained vision-language pre-training. By using a token-wise maximum similarity, our method learns fine-grained representation for patches in the images and words in the sentences. While it achieves competitive results against several large-scale multi-modal pre-training on various downstream tasks, both its architecture and training procedure can still be optimized to improve its performance. In the future, a more advanced image encoder as well as a well-designed interaction layer can be used to boost the performance. Furthermore, we can further add more masked language/image loss to support more generation tasks. To this end, we hope to extend FILIP as a generic and unified interface for solving a large variety of vision-language tasks.

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

## A  APPENDIX

### A.1  ETHICAL ISSUES IN DATA COLLECTION

When collecting the large-scale image-text pairs from the Internet, we perform person-name substitutions as in Changpinyo et al. (2021), in order to protect the privacy of the individuals appearing in the text. Specifically, we replace each person name appeared in the text with a special $<person>$ token. Besides, we also disgard image-text pairs whose text contains sensitive words.

### A.2  DATASETS SUMMARY

Table 6 shows the number of image-text pairs of each datasets used in different pre-training methods.

Table 6: Number of image-text pairs used in the pre-training of FILIP, CLIP and ALIGN.

|  | FILIP | | | | CLIP | ALIGN |
|---|---|---|---|---|---|---|
|  | CC3M | CC12M | YFCC100M | FILIP300M | (Radford et al., 2021) | (Jia et al., 2021) |
| # | 3M | 10M | 26M | 300M | 400M | 1800M |

### A.3  DETAILED EXPERIMENTAL SETTINGS

Table 7: The architecture parameters for FILIP models.

| Model | Embedding dimension | Input resolution | Image Encoder | | | Text Encoder | | |
|---|---|---|---|---|---|---|---|---|
|  |  |  | #layers | width | #heads | #layers | width | #heads |
| FILIP$_{base}$ | 256 | $224 \times 224$ | 12 | 768 | 12 | 12 | 512 | 8 |
| FILIP$_{large}$ | 256 | $224 \times 224$ | 24 | 1024 | 16 | 12 | 768 | 12 |

**Model Architectures.** We follow the same architecture design as CLIP, for both FILIP$_{base}$ and FILIP$_{large}$, except that we reduce the embedding dimension from 512/768 to 256 for the efficiency of loss computation. Table 7 describes the details of architectures.

**Details for Pre-training and Hyperparameters.** For the implementation of the contrastive loss, following CLIP (Radford et al., 2021) and ALIGN (Jia et al., 2021), we also set the temperature in the softmax function to be a learnable parameter and initialize it as 0.07. For the pre-training, we use the LAMB optimizer implemented by the cybertronai's open-source repository (https://github.com/cybertronai/pytorch-lamb). For the learning rate scheduler, we first assign a base learning rate and then linearly warm it up to the peak learning rate according to the effective total batch size by a square root strategy, $peak\_lr = base\_lr \times \sqrt{\frac{total\_bs}{512}}$. We note that a large weight decay is crucial to stabilize training and improve generalization. Specifically, we found that the training stability is a challenging issue when applying mix-precision training to large-scale models, i.e., the training is extremely unstable and the NaN loss easily happens. Recent works

Table 8: Common hyperparameters used for FILIP pre-training.

| Hyperparameter | Value |
|---|---|
| Vocabulary size | 49408 |
| Initial temperature | 0.07 |
| LAMB $\beta_1$ | 0.9 |
| LAMB $\beta_2$ | 0.999 |
| LAMB $\epsilon$ | $10^{-4}$ |
| Warm-up iters | 3000 |
| Training epochs | 30 |

Table 9: Model- and dataset-specific hyperparameters used for FILIP pre-training. Numbers in batch size represent the total batch size across all workers and are calculated as: batch size per GPU $\times$ #GPUs. FILIP340M is the combination of FILIP300M, YFCC100M, CC12M and CC3M.

| Model | Dataset | Batch size | Base LR | Weight decay |
|-------|---------|------------|---------|--------------|
| FILIP$_{base}$ | YFCC100M | $1024 \times 8$ | $6 \times 10^{-3}$ | 3e-2 |
| FILIP$_{base}$ | FILIP340M | $320 \times 128$ | $2 \times 10^{-3}$ | 3e-3 |
| FILIP$_{large}$ | FILIP340M | $160 \times 192$ | $1.5 \times 10^{-3}$ | 3e-3 |

DALL-E (Ramesh et al., 2021) and Cogview (Ding et al., 2021) also notice this issue and provide their solutions. However, we found that simply increasing the weight decay and applying the trick of removing the weight decay of specific parameters as described in Section 4.1 work for our case. The base learning rate and weight decay are selected manually via observing the performance at the early training stage. Table 8 summarizes the common hyperparameters and Table 9 shows the model- and dataset-specific hyperparameters for FILIP pre-training.

**Details for Image-text Retrieval.** Following previous works (Jia et al., 2021; Li et al., 2021a), for Flickr30K, we test on the 1K test set with or without fine-tuning on the 30K training set, while for MSCOCO, we test on the 5K test set with or without fine-tuning on the 113K training set. We use the similarity between image and text for ranking and use the contrastive loss for fine-tuning. Since there are multiple texts for each image in these two datasets, we change the ground-truth label of contrastive loss to consider multiple positives, by assigning a probability of 1/#positive to each positive following ALBEF (Li et al., 2021a). Besides, we also use prompts during evaluation for both datasets, see Appendix A.5 for details. Table 10 shows the hyperparameters for image-text retrieval fine-tuning.

Table 10: Hyperparameters used for image-text retrieval fine-tuning.

| Hyperparameter | Value |
|----------------|-------|
| Image size | $392 \times 392$ |
| Training epochs | 3 |
| Optimizer | LAMB |
| Batch size | 5120 |
| Base LR | $2 \times 10^{-4}$ |
| Weight decay | $3 \times 10^{-4}$ |

## A.4 MORE VISUALIZATIONS OF WORD-PATCH ALIGNMENT AND GRAD-CAM HEATMAPS

In Figure 3, we visualize the cross-modal alignment of the proposed method for more images, in terms of both word-patch alignment as described in Section 4.5 and Grad-CAM heatmaps (Selvaraju et al., 2017). We compute the Grad-CAM heatmaps based on the average self-attention maps over the image patches classified to targeted textual tokens (i.e., the textual token(s) corresponding to the class label in the ImageNet dataset) in the last layer of the image encoder. We average the heatmaps over all attention heads. As can be seen, our proposed model learns meaningful alignment between image patches and textual tokens.

## A.5 PROMPT TEMPLATES FOR DOWNSTREAM TASKS

**Image Classification.** Table 11 shows the prompt templates for different image classification datasets in the form of " [prefix] {label}, [category description]. [suffix]. " in Equation (6). There are three components to be determined in the template, i.e., the prefix, the category description and the suffix. For each component, we select several well-performed ones for each dataset. Then we use the full combinations of all three components as the set of prompt templates for ensemble. For instance, we use 5 prefixes, no category descriptions, and 6 suffixes for dataset ImageNet. Then the total number of prompt templates for this dataset is: $5 \times 1 \times 6 = 30$.

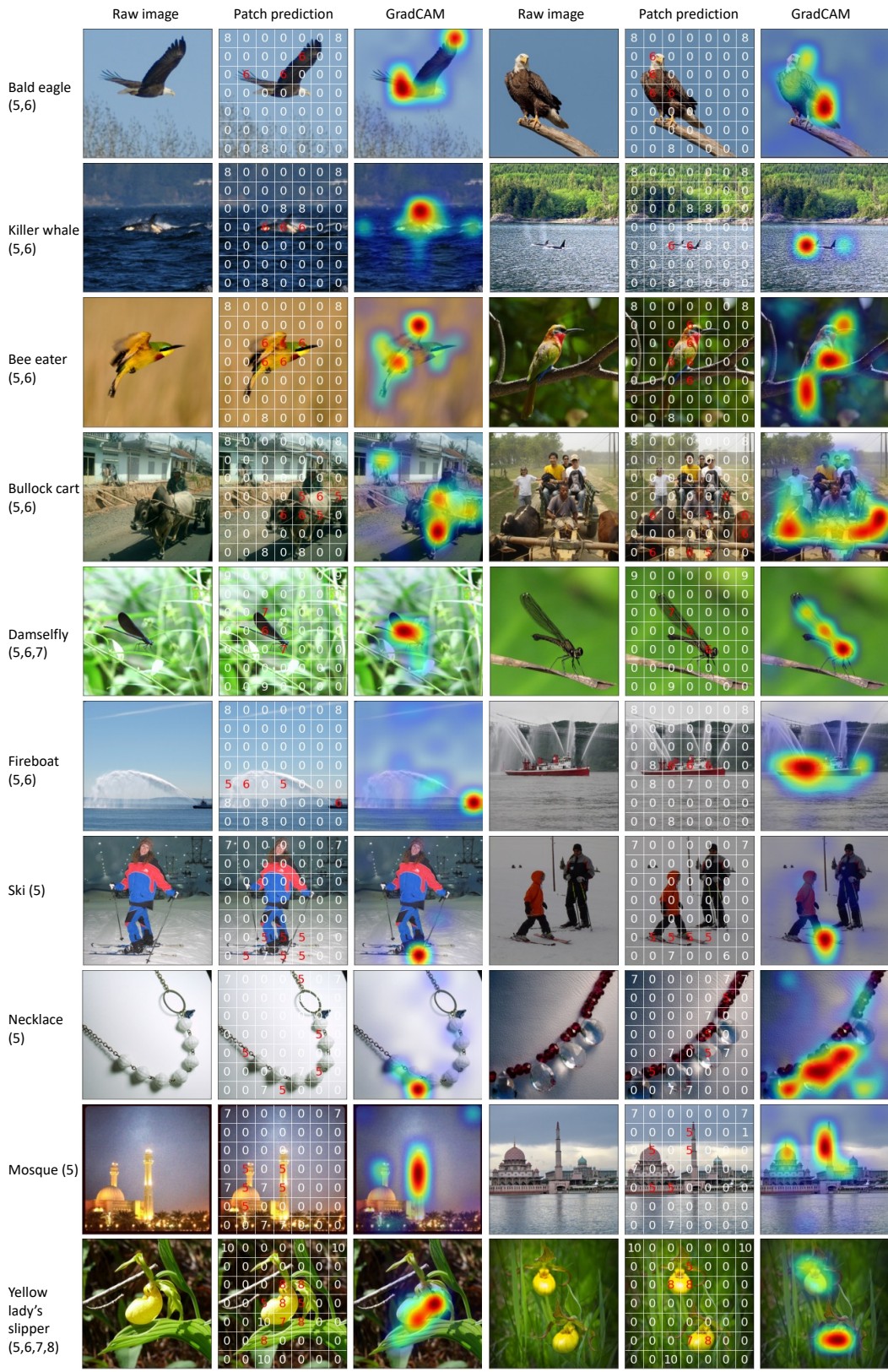

Figure 3: More visualizations on different classes of ImageNet dataset. Numbers in the parentheses after the class label indicate the location indices of class label in the tokenized textual sequence.

Table 11: Prompt templates used for 12 downstream image classification tasks.

| Dataset | Prefix | Category description | Suffix |
|---------|--------|---------------------|--------|
| CIFAR10 | "a photo of a", "a jpeg photo of a", "a painting of a", "itap of a", "graffiti of a", "a cartoon", "a doodle" | None | None, "It's common in daily life", "It's cute", "It's ugly", "It's weird", "Hope you like it" |
| CIFAR100 | "a jpeg photo of a", "a painting of a", "a good photo of a", "a bad photo of a", "a photo of a", "itap of a", "a rendering of a" | None | None, "It's common in daily life", "It's beautiful", "It's ugly", "I like it", "I take it today" |
| Caltech101 | "a photo of a", "a cropped photo of a", "a good photo of a", "a bad photo of a" | None | None, "I like it", "I hate it", "It's ugly", "It's cute" |
| Stanford-Car | "a photo of a", "a close-up photo of a", "a good photo of a", "a bad photo of a" | "a type of car", "a type of automobile" | "I like it", "It belongs to my friend", "It's brand new", "It's popular recently", "It's important to me", "I take it today" |
| Flowers102 | "a photo of a (many) ", "a rendering of a (many) ", "itap of a (many) " | "a type of flower", "a type of bloom" | "It's beautiful", "It's from my best friend", "It gives out a sweet perfume/fragrance" |
| ImageNet | "a photo of a", "a good photo of a", "a bad photo of a", "a close-up photo of a", "itap of a" | None | "I like it", "It's common in daily life", "It's not common in daily life", "It's ugly", "It's cute", "It's beautiful" |
| Food101 | "a photo of my", "a close-up photo of my", "itap of my" | "a type of food", "a type of nourishment" | "I made it today", "I like it", "I hate it", "It's delicious", "It's with nice flavour", "It's with terrible flavour", "It's popular recently" |
| SUN397 | "a photo of a", "a good photo of a", "a bad photo of a", "a bright photo of a", a dark photo of a", "a black and white photo of a", "a nice scene of a", "a terrible scene of a" | None | None, "I like it", "I hate it", "It's beautiful", "It's common in daily life", "It's important to me" |
| DTD | "itap of a", "a close-up photo of a" | "texture", "surface", "material" | None, "It's out of style", "It's popular in old days", "It's ugly", "It's beautiful" |
| Aircrafts | "a photo of the", "a close-up photo of the", "a good photo of the ", "a pixelated photo of the" | "a type of plane", "a type of aircraft", "a type of airliner" | None,"I like it", "It's important to me", "I take it today", "Hope you like it" |
| Oxford Pet | "a photo of my", "a low resolution photo of my", "a good photo of my" | "a type of pet", "a type of dog or cat" | None, "It's cute", "It's important to me", "I like it", "It's beautiful" |
| EuroSAT | "a photo of a", "a painting of a", "a cropped photo of a", "a good photo of a", "a blurry photo of a" | None, "an example of aerial or satellite images" | None, "I like it", "It's taken from an aircraft or some flying object", "It's collected by imaging satellites" |

Table 12: Prompt templates used for zero-shot image-text retrieval on Flickr30K and MSCOCO datasets.

| Dataset | Task | Prefix | Suffix |
|---------|------|--------|--------|
| Flickr30K | image-to-text retrieval
text-to-image retrieval | "a good photo of the"
"a good photo of" | "I hate it."
None |
| MSCOCO | image-to-text retrieval
text-to-image retrieval | "a good photo of"
None | "It is ugly."
None |

**Image-text Retrieval.** Following CLIP (Radford et al., 2021), we use prompt in zero-shot image-text retrieval for both Flickr30K and MSCOCO datasets. The prompt is selected by the same rule as described in Section 3.1.2, except that we do not use "[category description]" here. Table 12 shows the prompt templates for zero-shot image-text retrieval on Flickr30K and MSCOCO datasets.

A.6 Linear Probe on Image Classification

In this section, we evaluate FILIP on the linear probe for image classification. Following common linear probe setting, we freeze the whole backbone network and only finetune the last linear classifier. Since we remove the "[CLS]" token in our vision encoder, we apply a mean pooling over all the other visual tokens to aggregate them into a global image representation which is then fed into the linear classifier.

**Setting.** Following CLIP, we train the logistic regression classifier using scikit-learn's L-BFGS implementation (Pedregosa et al., 2011), with maximum 1,000 iterations on those 11 datasets except ImageNet. For ImageNet, we use a pytorch-based codebase to accelerate the training with GPU. Following Doersch et al. (2015), we adopt a Batch Normalization (Ioffe & Szegedy, 2015) layer before the linear classifier which is beneficial to stabilize the mixed-precision training. Random resized crop and horizontal flipping are used to augment training data. We use the cosine learning rate scheduler with a linear warmup of 10 epochs. More hyperparameters used in linear probe on ImageNet are shown in Table 13.

Table 13: Hyperparameters used for linear probe image classification on ImageNet.

| Hyperparameter | Value |
|----------------|-------|
| Image size | $224 \times 224$ |
| Training epochs | 90 |
| Optimizer | SGD |
| Batch size | 4096 |
| Base LR | 0.1 |
| Weight decay | 0 |

**Results.** Table 14 compares the linear probe performance of our proposed FILIP with CLIP over 12 datasets. Our FILIP$_{base}$ (resp. FILIP$_{large}$) achieves 85.5% (resp. 91.0%) average Top-1 accuracy over 12 downstream tasks, which provides noticeable improvements, i.e., 1.8% (resp. 1.2%) higher, compared to its CLIP's counterpart. This implies that our FILIP learns more powerful vision features which may potentially facilitate border downstream vision tasks.

A.7 Comparison with Khattab & Zaharia (2020)

As is stated in Section 3.1, compared to Khattab & Zaharia (2020), besides being the first to apply the late interaction to contrastive learning for vision-language pre-training, we make two other modifications, i.e., removing padded tokens and using average over non-padded tokens instead of summation. In the following, we show that these two modifications are crucial to the performance, and the quality of finer-granular word-patch alignment.

For comparison, we replace the proposed cross-modal late interaction in FILIP$_{base}$ with the original late interaction in Khattab & Zaharia (2020). Following the setting in Section 4.4, we pre-train on

Table 14: Top-1 accuracy(%) of linear probe on image classification on 12 datasets. Our FILIP outperforms CLIP by 1.2∼1.8% points on average.

| | CIFAR10 | CIFAR100 | Caltech101 | StanfordCars | Flowers102 | Food101 | SUN397 | DTD | Aircrafts | OxfordPets | EuroSAT | ImageNet | Average |
|---|---|---|---|---|---|---|---|---|---|---|---|---|---|
| CLIP-ViT-B/32 | 95.1 | 80.5 | 93.0 | 81.8 | 96.6 | 88.8 | 76.6 | 76.5 | 52.0 | 90.0 | 97.0 | 76.1 | 83.7 |
| FILIP$_{base}$ | 95.3 | 80.3 | 95.0 | 78.6 | 98.7 | 86.2 | 77.9 | 78.1 | 76.6 | 88.0 | 95.9 | 75.8 | **85.5**$^{+1.8}$ |
| CLIP-ViT-L/14 | 98.0 | 87.5 | 96.5 | 90.9 | 99.2 | 95.2 | 81.8 | 82.1 | 69.4 | 95.1 | 98.2 | 83.9 | 89.8 |
| FILIP$_{large}$ | 97.9 | 87.0 | 97.2 | 89.0 | 99.6 | 94.6 | 83.2 | 83.9 | 84.8 | 93.5 | 97.3 | 84.5 | **91.0**$^{+1.2}$ |

the filtered YFCC100M with mixed-precision using 8 V100 GPUs. The batch size per GPU is 512 and the dimension of the token feature is 256. We report results with the top 25% tokens (selected using the method in Section 3.1) during training. Note that the original late interaction in Khattab & Zaharia (2020) is sensitive to the temperature in the softmax function, and we report the best result among several initialization values of the temperature.

**Effect to Performance.** Table 15 shows the comparison on zero-shot ImageNet classification. When these two modifications are removed, the zero-shot Top-1 accuracy of ImageNet drops from 34.3 to 32.7.

Table 15: Comparison of Top-1 Accuracy(%) between the proposed cross-modal late interaction loss and Khattab & Zaharia (2020) on zero-shot ImageNet classification.

| ours | late interaction in Khattab & Zaharia (2020) |
|---|---|
| **34.3** | 32.7 |

**Effect to the Word-patch Alignment** In Figure 4, we compare the word-patch alignment using the models trained with the proposed cross-modal late interaction and the late interaction in Khattab & Zaharia (2020). According to the visualizations, using the original late interaction in Khattab & Zaharia (2020) leads to less accurate word-patch alignment. Specifically, the object patches are often aligned to the padded tokens instead of class names. We speculate this is because the padded tokens learn similar representations as existing key textual tokens, similar to the finding in Section 3.2 of Khattab & Zaharia (2020) that padding with masked tokens (which is called "query augmentation" in Khattab & Zaharia (2020)) tend to "re-weigh existing terms based on their importance for matching the query".

A.8    ABLATION ON THE FULL PRE-TRAINING DATASET

In Table 16, we compare the proposed cross-modal late interaction loss with the original CLIP loss (Radford et al., 2019) on the full pre-training dataset introduced in Section 3.3. In Table 16, CLIP denotes the results reported by CLIP paper, CLIP$_{rep}$ is our reproduced CLIP version with the original contrastive loss using exactly the same architecture on the same pre-training dataset as FILIP$_{base}$. As can be seen, the FILIP$_{base}$ has 6.7 points higher average accuracy than the CLIP$_{rep}$ over 12 datasets. This further verifies that the performance gain of FILIP comes from the proposed cross-modal late interaction, rather than the data or architecture.

A.9    INFERENCE TIME OF IMAGE-TEXT RETRIEVAL

**Setting.** In this section, we test the inference time of both image retrieval and text retrieval on the test set of Flickr30K and MSCOCO. We compare our proposed model FILIP$_{large}$ against SCAN (Lee et al., 2018) and CLIP (ViT-L/14) (Radford et al., 2021) . We test the inference time of CLIP and SCAN using their released code. For image retrieval, we precompute the image features and report the inference time for one text query, which contains (i) the time to extract the feature of one text query, and (ii) the time of similarity calculation with all images and ranking. Similarly, for

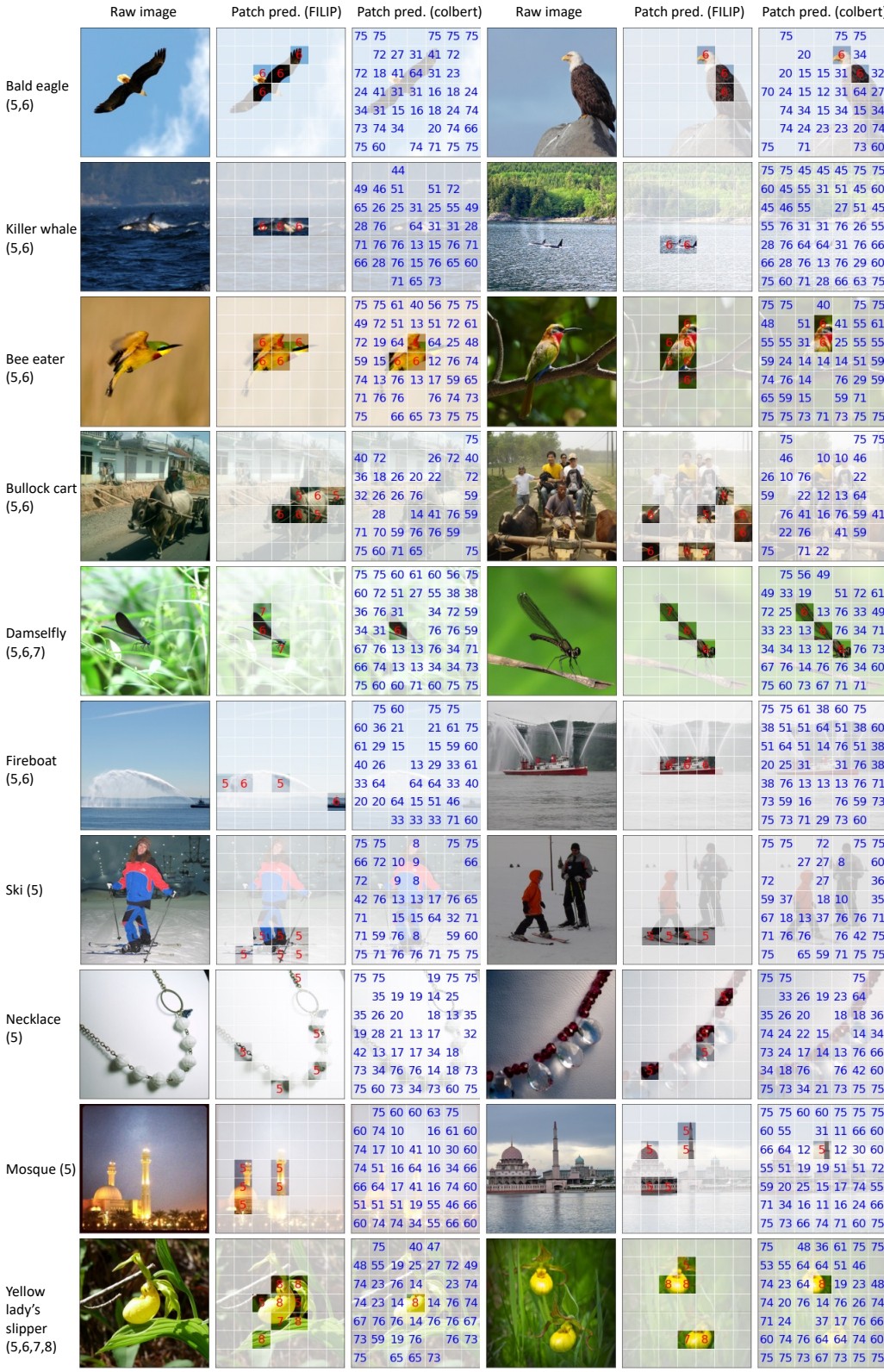

Figure 4: Comparison of word-patch alignment between the proposed cross-modal late interaction and that in ColBERT (Khattab & Zaharia, 2020). "a photo of a {label}." is the prompt. Numbers in the parentheses after the class label indicate the location indices of the class label in the tokenized textual sequence. The correct predictions to the class labels are highlighted by opaque patches with the class label indices in red. Incorrect predictions to the padded tokens are highlighted by opaque patches with the padded token indices in blue.

Table 16: Top-1 accuracy(%) on image classification on 12 datasets. $\text{CLIP}_{\text{rep}}$ is our reproduced CLIP trained with the same training data and evaluated with the same prompts as our FILIP. With the same backbone architecture, our FILIP significantly improves the zero-shot Top-1 average accuracy over 12 datasets.

| | CIFAR10 | CIFAR100 | Caltech101 | StanfordCars | Flowers102 | Food101 | SUN397 | DTD | Aircrafts | OxfordPets | EuroSAT | ImageNet | Average |
|---|---|---|---|---|---|---|---|---|---|---|---|---|---|
| CLIP | 91.3 | 65.1 | 87.9 | 59.4 | 66.7 | 84.4 | 63.2 | 44.5 | 21.2 | 87 | 49.4 | 63.2 | 65.3 |
| $\text{CLIP}_{\text{rep}}$ | 82.0 | 57.5 | 89.9 | 45.1 | 80.7 | 75.1 | 63.6 | 46.7 | 33.7 | 82.7 | 49.0 | 64.2 | 64.2 |
| $\text{FILIP}_{\text{base}}$ | 86.9 | 65.5 | 91.9 | 55.4 | 85.3 | 82.8 | 69.1 | 49.3 | 57.2 | 88.1 | 49.9 | 68.8 | **70.9** |

Table 17: Comparison on performance and inference time of image-text retrieval on Flickr30K and MSCOCO datasets.

| | | | Recall | | | | | | | | | | Inference time | | | |
|---|---|---|---|---|---|---|---|---|---|---|---|---|---|---|---|---|
| | | Flickr30K | | | | | | MSCOCO | | | | | Flickr30K | | MSCOCO | |
| | image->text | | | text->image | | | image->text | | | text->image | | | | | | |
| | R@1 | R@5 | R@10 | R@1 | R@5 | R@10 | R@1 | R@5 | R@10 | R@1 | R@5 | R@10 | i-to-t | t-to-i | i-to-t | t-to-i |
| SCAN | 67.4 | 90.3 | 95.8 | 48.6 | 77.7 | 85.2 | 50.4 | 82.2 | 90.0 | 38.6 | 69.3 | 80.4 | 4.47s | 7ms | 21.3s | 26ms |
| CLIP | 88.0 | 98.7 | 99.4 | 68.7 | 90.6 | 95.2 | 58.4 | 81.5 | 88.1 | 37.8 | 62.4 | 72.2 | 23ms | 8ms | 24ms | 9ms |
| FILIP | 96.6 | 100.0 | 100.0 | 87.1 | 97.7 | 99.1 | 78.9 | 94.4 | 97.4 | 61.2 | 84.3 | 90.6 | 24ms | 8ms | 26ms | 9ms |

text retrieval, we precompute the text features and report the inference time for one image query, which contains (i) the time to extract the feature of one image query, and (ii) the time of similarity calculation with all texts and ranking. The test set of Flickr30k contains 1000 images and 5000 texts, while the test set of COCO contains 5000 images and 25000 texts. The time is averaged over 1000 runs.

**Results.**    The inference time of retrieval is shown in Table 17. Benefitting from the efficiency optimizations (i.e., FP16 quantization and reduced feature dimension) in Section 3.1, the inference time of FILIP is close to CLIP. In image retrieval, SCAN is slightly faster than FILIP on Flickr30K with 1000 images, because SCAN uses a lightweight GRU as the text encoder. However, SCAN is much slower than FILIP (i.e., about 17ms slower per query) on MSCOCO with more (i.e., 5000) images because of the slower computation involved in the two-stage stacked cross-attention when computing the similarity. For text retrieval, SCAN is much slower than FILIP and its own image retrieval, mainly due to three reasons: (i) the image encoder is a Faster RCNN which is more expensive than the lightweight GRU text encoder; (ii) the text candidates are 5 times more than the image candidates in image retrieval; and (ii) the similarity computation of SCAN relies on the cross-attention computation, which is not straightforward to be paralleled, even in their official code; while our FILIP's similarity computation is simply a matrix multiplication and is readily optimized on most modern hardwares.

