# OpenReview forum: "FILIP: Fine-grained Interactive Language-Image Pre-Training"
_ICLR.cc/2022/Conference — ICLR 2022 Poster_

### Official Review · Reviewer_AMdt · 2021-11-01

**Correctness:** 3
**Technical Novelty And Significance:** 2
**Empirical Novelty And Significance:** 3
**Recommendation:** 6
**Confidence:** 4

**Main Review:**

Strengths:
* The paper is well written and the assumptions made by the authors are well motivated and described with respect to the state of the art and literature.
* The model is quite simple and easy to implement when compared to UNITER and Unicoder-VL.
* The results of the experiments consistently show improvement wrt other state-of-the-art methods.

On the fence (to discuss):
1. Novelty is quite limited given that it seems to be a combination of existing models.
   1. The use of ViT in the vision-language pre-training setup is quite interesting, thought it is not the main novelty since it has been used in other vision-language pre-training models (e.g., CLIP and ALIGN).
   2. The most interesting contribution is to do alignment via late fusion following the method in Khattab & Zaharia (2020). The authors explain the differences wrt that paper at the end of page 4 and beginning of page 5. However, they are quite limited: using average instead of sum, excluding padded tokens, and change to contrastive loss.
   3. Although each single piece taken independently is not very novel, I think there is some value on the combination especially in such applications.

2. I understand that most of these new methods are trained on a very large scale dataset scraped from the web and it is hard to share them. At the same time, it is impressive to see that the authors have created their own cleaner version of such pre-training datasets. However, I did not see a single signal in the paper, that the authors plan to release the data or links to rebuild the data or even code. This limits the reproducibility of the experiments and also it lowers the trust on the results. I'd suggest the authors to argument on this topic and possibly commit on sharing any of the resources to being able to reproduce experiments and compare with their method.

3. It is unclear why the authors run the ablation studies using YFCC100M for pre-training and not FILIP100M. It is appreciated that there is a thorough ablation study on YFCC100M. But why wasn't it carry out on FILIP100M?

Weaknesses:

4. Computational complexity at inference time is not clear. It seems that one can lower the computation burden of training by just using heuristics, such as keeping 25% of tokens. It makes absolutely sense to be able to do this during training, because of different epochs, randomization and large dataset. However, it is not clear what happens at test time:
   1. Does the method extract 25% of tokens of the query and the data in the search index?
   2. If yes to 4.1, which tokens should be chosen for the query (can be done on the fly) and which one for the search index (it should be fixed for computational reasons)?
   3. If no to 4.2, then the claim that the method is more efficient is not valid anymore, because the comparisons to made are still quadratic as in self-attention. Anything that I am missing that makes the method more efficient for a different reason?
   4. If we need to store the full fine-grained tokes in a search index, this means that we will have 77 tokens (as stated in the experiment section) compared to the 2 for CLIP-based methods. This means that the index is way bigger than the original one, if we were to use global features. Is this thinking correct? If yes, any idea on how to make this better?

Other improvements
* Usage of the word "Interactive". I am not sure why the authors used this word in the title and in other places in the manuscript. I do not see how the method requires interaction from a person or user. To me it is more about correlation or alignment between image and text.
* Eq 3, 4, 5 are a bit confusing. They can be simplified as did in the paper of Khattab & Zaharia (2020), where you use the max(\cdot) in eq 3 directly inside eq 4 and 5. So you do not need eq 3.


**Summary Of The Paper:**

The paper tackles the problem of large-scale vision-language pre-training which has been quite popular recently. The main issues identified by the paper are: 1) fine-grained information might be lost due to the use of global features, and 2) alignment between text and language is typically done using cross- or self-attention, which are inefficient. The authors proposed a dual-stream method that 1) uses visual "patch" tokens from ViT models and 2) perform late fusion of the two modalities using a variant of the method from Khattab & Zaharia (2020).

**Summary Of The Review:**

The authors do a good job on putting together ideas from different papers and create a novel pre-training dataset which leads to new state of the art results on vision-language pre-training tasks. However, the contributions seem to be marginally significant, and I'd like to hear more from the authors about the questions I raised above. Moreover, the computational complexity claim needs to be revised and discussed more in depth, since the high price for storing the index might not compensate the small improvement in accuracy performance. If any of those two points are covered in the discussion phase, I am willing to reevaluate my score recommendation.

---

> ### Author Response · Authors · 2021-11-21
> **Reply to Reviewer 4**
>
> **Q1: Novelty is quite limited given that it seems to be a combination of existing models. The use of ViT in the vision-language pre-training setup is quite interesting, thought it is not the main novelty since it has been used in other vision-language pre-training models (e.g., CLIP and ALIGN).**
>
> **Reply:** (1) Our key contribution of this work does not lie in the dual-stream model architecture (including the use of VIT) itself, but lies in making  cross-modal late interaction tractable in contrastive learning for vision-language pre-training, to boost the downstream tasks' performance and simultaneously allows fine-grained word-patch alignment, which is not explored in previous works.
>
> (2) Please also refer to our reply to the Q1 of Reviewer 1, where we discussed in detail about the novelty of this work.
>
> **Q2: The most interesting contribution is to do alignment via late fusion following the method in Khattab \& Zaharia (2020). The authors explain the differences wrt that paper at the end of page 4 and beginning of page 5. However, they are quite limited: using average instead of sum, excluding padded tokens, and change to contrastive loss. Although each single piece taken independently is not very novel, I think there is some value on the combination especially in such applications.**
>
> **Reply:** (1) Thanks. As is said by the reviewer, the main novelty of this work lies in making  cross-modal late interaction tractable in contrastive learning for vision-language pre-training, to boost the downstream tasks' performance and simultaneously allow fine-grained word-patch alignment, which is not explored in previous works.
>
> (2) For the differences  between this work and Khattab \& Zaharia (2020), please refer to the last two points of our reply to the Q1 of Reviewer 1.
>
> **Q3: I understand that most of these new methods are trained on a very large scale dataset scraped from the web and it is hard to share them. At the same time, it is impressive to see that the authors have created their own cleaner version of such pre-training datasets. However, I did not see a single signal in the paper, that the authors plan to release the data or links to rebuild the data or even code. This limits the reproducibility of the experiments and also it lowers the trust on the results. I'd suggest the authors to argument on this topic and possibly commit on sharing any of the resources to being able to reproduce experiments and compare with their method.**
>
> **Reply:** Thanks for the suggestion. We plan to release a dataset with hundreds of millions image-text pairs and the inference code if the paper is accepted, for easier comparison with other methods and development of this community.
> Furthermore, as we understand the large-scale pre-training may be too costly for colleges and individual researchers, we plan to release some pre-trained models for benchmarking the performance of large-scale multi-modal models as well as  facilitating more research on downstream tasks and applications.
>
> **Q4: It is unclear why the authors run the ablation studies using YFCC100M for pre-training and not FILIP100M. It is appreciated that there is a thorough ablation study on YFCC100M. But why wasn't it carry out on FILIP100M?**
>
> **Reply:** We carry out the ablation on the filtered YFCC100M due to several reasons:
> (1) YFCC100M is publicly available and doing ablation on it is fairer.
> (2) CLIP also reports performances using a filtered YFCC100M. Experimenting on a similar dataset makes us better compare with them, and it is also conducive for us to verify our models' performance.
> (3) YFCC100M is suitable for ablation in both quality and volume. It contains only ~26M image-text pairs while still maintains a good diversity, which is better than roughly sampling a subset from a large dataset. For example, CLIP has tried to experiment on a subset of their WIT dataset, but it results in a 3.7\% drop in zero-shot ImageNet Top1 accuracy compared to using the YFCC with similar volume (see Table 12 in CLIP's paper).
>
> **Q5: Computational complexity at inference time is not clear. It seems that one can lower the computation burden of training by just using heuristics, such as keeping 25\% of tokens. It makes absolutely sense to be able to do this during training, because of different epochs, randomization and large dataset. However, it is not clear what happens at test time: Does the method extract 25\% of tokens of the query and the data in the search index?**
>
> **Reply:** We do not use 25\% tokens during inference.

---

> > ### Author Response · Authors · 2021-11-21
> > **Reply to Reviewer 4**
> >
> > **Q6:  If no to the Q5, then the claim that the method is more efficient is not valid anymore, because the comparisons to made are still quadratic as in self-attention. Anything that I am missing that makes the method more efficient for a different reason?**
> >
> > **Reply:** (1) Though both are quadratic to sequence length, computing only the token-wise similarity in FILIP can be several magnitudes cheaper than running the model with self-attention. This is because in a $L$-layer Transformer-based model, the Feed-Forward-Network (FFN) is often more expensive than the self-attention especially when the sequence length is small, and both self-attention and FFN need to be computed $L$ times.
> >
> > (2) Specifically, we use Floating Point Operations (FLOPs) to measure the computation complexity.
> > (i) For computing the token-wise similarity in $\rm FILIP_{base}$, the FLOPs required  between $n_1=77$ textual tokens and $n_2=49$ patches with feature dimension $d_1=256$ is $2n_1n_2d_1=2 \times (49 \times 77 \times 256)=1.9M$. (ii) On the other hand, the FLOPs required for running a  sequence with length $n=n_1+n2=126$ through a 12-layer (i.e., $L=12$) VIT-B/32 (the same size as the image encoder of $\rm FILIP_{base}$) with hidden dimension $d=768$ is over $2L(2n^2d+4nd^2+2nd \times 4d)=22.0G$ for only the matrix multiplications involved, which is 4 magnitudes more expensive than computing the token-wise similarity in $\rm FILIP_{base}$.
> >
> > **Q7: If we need to store the full fine-grained tokens in a search index, this means that we will have 77 tokens (as stated in the experiment section) compared to the 2 for CLIP-based methods. This means that the index is way bigger than the original one, if we were to use global features. Is this thinking correct? If yes, any idea on how to make this better?**
> >
> > **Reply:** (1) The reviewer is correct that FILIP need to store token-wise representations. Note that this is not a big issue for the classification tasks for the small-scale retrieval tasks studied in the paper, because the number of samples need to be stored is relatively small.
> >
> > (2) For usage in a real retrieval system with large search index, as is mentioned at the end of Section 3.1.1, we use the FP16 feature instead of FP32 ones, and reduce the feature dimension from the original 768/512 to 256. These two techniques save the storage by at least 4 times. We are also trying to further quantize the features to int8/int4, as well as token-pruning methods to further reduce the storage burden.
> >
> > #### Minor
> >
> > **Q8: The word "fine-grained Interactive" in the title highlights the interaction between image patches and words in the text. Note that previous SOTA methods such as CLIP and ALIGN cannot learn meaningful fine-grained patch features with promising localization ability as our FILIP.**
> >
> > **Reply:** The word "fine-grained Interactive" in the title highlights the interaction between image patches and words in the text. Note that previous SOTA methods such as CLIP and ALIGN cannot learn meaningful fine-grained patch features with promising localization ability as our FILIP.
> >
> > **Q9: Eq 3, 4, 5 are a bit confusing. They can be simplified as did in the paper of Khattab \& Zaharia (2020), where you use the max in eq 3 directly inside eq 4 and 5. So you do not need eq 3.**
> >
> > **Reply:** We agree that the reviewer that using a max in eq 4,5 can make the formulation look simpler. We choose to write Eq 3 out in order to explain the intuition behind the cross-modal late interaction in Remark 1.

---

> > > ### Comment · Reviewer_AMdt · 2021-11-29
> > > **Reply to the Rebuttal**
> > >
> > > Thanks for the insightful answers. I really appreciated that the authors covered all my points with a good level of details. I remain a bit puzzled about the computational cost to store the features, but I fully understand your points.

---

### Official Review · Reviewer_gJkL · 2021-11-02

**Correctness:** 3
**Technical Novelty And Significance:** 2
**Empirical Novelty And Significance:** 2
**Recommendation:** 5
**Confidence:** 4

**Main Review:**

Strengths:
(1) The proposed FILIP method achieve SoTA performance  on ITM task, which is significant for this area.
(2) FILIP300M is a dataset of 300 million image-text paired samples collected and cleaned from the Internet, which is a huge amount of work, and will be beneficial to the whole V-L community.


Weaknesses:
(1) The core idea of FILIP method, namely to achieve word-patch alignment by token-wise similarity matrix through cross-modal late interaction, as far as we know, is first originated from the classical ITM method SCAN [1] (see below). So, FILIP’s cross-modal late interaction by Eqs. (3)(4)(5) is only equivalent to of SCAN’s Sum-Max (without attention) by Eqs. (9)(10), which is only a special case in SCAN’s ablation studies (see Table 3). Therefore, FILIP is lack of novelty, and we suggest the authors perform more comparative experiments about various pooling/attention/scoring methods different from SCAN [1].

(2) FILIP is not efficient in similarity matching stage, althougn it is efficient in feature extraction and embedding stage. That is because FILIP is a special case of SCAN [1], and SCAN is time consuming in cosine similarity matching, which is a commonly recognized in V-L communities. So, when the test data are very huge (for example, the real image-text search engine, usually millions of image-text pairs in total, rather than 1K/5K test data), the similarity matching time will beyond endurance. So, we suggest the authors perform some comparative experiments on the 1K/5K testing speed/time on f30k/coco with other methods like CLIP, SCAN [1] to prove their efficiency on inference phase.

(3) The large-scale V-L pre-training dataset are not public/available. This might not be a very big problem since many other similar works also did this. But we cannot sure whether there are similar images/texts between the large-scale pre-training dataset (training + validation split) and the f30k/coco baseline dataset (val/test split). If the answer is true, then it is definitely unfair.

(4) The SoTA performance are mainly come from 2 aspects:
(i) The FILIP300M large-scale pre-training dataset and its cleaning/pre-processing steps.
(ii) Grid/Patch feature (1 image, multiple features) methods like ViTs, which are the latest advances of backbone networks in CV.
So, it’s SoTA performance should not be largely due to the proposed cross-modal late interaction.

(5) ViTs (1 image, multiple features) have the common problem of low efficiency in training/inference speed than traditional CNNs (1 image, 1 global feature). So, this shortcoming may be Inherited by FILIP. So, the training/testing speed may not exceed CLIP/ALIGN. Can the authors give more explanations?

(6) The statements in this paper have a double standard on “offline/pre-computing”.
(i) For other works: “This design complicates ... due to pre-computing and storing a large number of ROI features” (page 1).
(ii) For their work：“FILIP successfully ... while simultaneously ... to pre-compute image and text representations offline” (page 1)
Therefore, we don’t know weather offline/pre-computing is efficient or not?

(7) There are some inappropriate expressions/mistakes in the introduction (page 1).
(i) “One line of work” can be substitute by “Bottom-up attention [2] (see below) methods”.
(ii) “Another line of work” can be replaced by “Cross-attention methods”.
(iii) “self-attention (Kim et al., 2021)” is inappropriate, because ViLT is a typical cross-attention method (see Figure 3, the cross-modal Transformer Encoder), rather than (pure) self-attention.

reference(s):
[1] Kuang-Huei Lee, Xi Chen, Gang Hua, Houdong Hu, and Xiaodong He. Stacked cross attention for image-text matching. In ECCV, 2018.
[2] Peter Anderson, Xiaodong He, Chris Buehler, Damien Teney, Mark Johnson, Stephen Gould, and Lei Zhang. Bottom-up and top-down attention for image captioning and visual question answering. In CVPR, 2018.


**Summary Of The Paper:**

(1) The core idea of the proposed FILIP method is to achieve word-patch alignment by token-wise similarity matrix through cross-modal late interaction by modifying only contrastive loss, which is both training and inference efficient. The authors collected FILIP300M, a large-scale cleaned image captioning dataset from the Internet for FILIP’s V-L pre-training.

(2) FILIP achieves SoTA performance on ZS image classification and ITM tasks. Visualization results show its promising ability of fine-grained (visual-textual token) classification and localization.

(3) To the best of our knowledge, FILIP is the first work with such high accuracy on F30k that not only does it reach double full mark of R@K (i.e. i2t[R@5 and R@10] are both 100.0), but also it achieves 580+ of rsum with only one single model (no ensemble).

**Summary Of The Review:**

The performance is very good, which mainly comes from  “big data + big model + data cleaning / augmentation + supercomputers”. The whole method is lack of novelty.

---

> ### Author Response · Authors · 2021-11-21
> **Reply to Reviewer 3**
>
> **Q1: The core idea of FILIP method, namely to achieve word-patch alignment by token-wise similarity matrix through cross-modal late interaction, as far as we know, is first originated from the classical ITM method SCAN [1] (see below). So, FILIP’s cross-modal late interaction by Eqs. (3)(4)(5) is only equivalent to of SCAN’s Sum-Max (without attention) by Eqs. (9)(10), which is only a special case in SCAN’s ablation studies (see Table 3). Therefore, FILIP is lack of novelty, and we suggest the authors perform more comparative experiments about various pooling/attention/scoring methods different from SCAN [1].**
>
> **Reply:** Thanks for pointing out this reference.  Our FILIP is different from SCAN in the following aspects:
>
> 1. SCAN relies on Faster-RCNN to locate and extract object features while our FILIP directly learns to localize fine-grained object from patch representation via ViT.
>
> 2. Our late interaction loss is based on contrastive learning which is scalable and can be applied on millions of image-text pairs. On the contrast, SCAN's alignment is based on Triplet loss with a bottom-Up attention. This diagram requires huge computations on extracting object features and the training scheme is not efficient. Their experiments are only done at scale of 100k training samples.
>
> 3. The performance gap is huge. E.g., for the performance of image-text retrieval: On flickr30k, our FILIP 96.6 R@1 VS SCAN 67.9; On MSCOCO, our FILIP 78.9 R@1 VS SCAN 50.4.
>
>
> As is stated in the introduction, our cross-modal late interaction modifies the similarity calculation, and does not rely on any attention computation like SCAN. Eqs (9) and (10) in SCAN indeed have the same sum-max formulation in Khattab \& Zaharia,(2020). For the difference with this max-sum token-wise interaction, we refer the reviewer to our reply to the first question from Reviewer 1  (especially the last two points).  Besides, we have added discussion on SCAN in related work.
>
> **Q2:  FILIP is not efficient in similarity matching stage, although it is efficient in feature extraction and embedding stage. That is because FILIP is a special case of SCAN [1], and SCAN is time consuming in cosine similarity matching, which is a commonly recognized in V-L communities. So, when the test data are very huge (for example, the real image-text search engine, usually millions of image-text pairs in total, rather than 1K/5K test data), the similarity matching time will beyond endurance. So, we suggest the authors perform some comparative experiments on the 1K/5K testing speed/time on f30k/coco with other methods like CLIP, SCAN [1] to prove their efficiency on inference phase.**
>
> **Reply:** (1) The reviewer is correct the token-wise similarity computation of FILIP can be slightly more expensive than CLIP which computes similarity with global features, but is much more efficient than attention-based single-stream methods like UNITER and OSCAR (refer to our reply to Q4 of Reviewer 2).
>
> (2) To improve the inference efficiency of FILIP, as is mentioned at the end of Section 3.1.1, we use the FP16 features instead of FP32 ones, and reduce the feature dimension from the original 512 to 256. We are also trying to quantize the features to int8/int4, as well as token-pruning methods to further reduce the storage burden.
>
> (3) In Appendix A.8, we compare the inference time of our proposed FILIP with CLIP and SCAN on image-text retrieval. Benefitting from the efficiency optimizations (i.e., FP16 quantization and reduced feature dimension) , the inference time of FILIP is close to CLIP.
> In image retrieval, SCAN is slightly faster than FILIP on Flickr30K with 1000 images, because SCAN uses a lightweight GRU as text encoder. However, SCAN is but much slower than FILIP (i.e., about 17ms slower per query) on MSCOCO with more (i.e., 5000) images because of the slow computation involved in the two-stage stacked cross-attention when computing the similarity. For text retrieval, SCAN is much slower than FILIP and its own image retrieval, mainly due to three reasons:
>
> (i) the image encoder is a faster R-CNN which is more expensive than the lightweight GRU text encoder; (ii) the text candidates are 5 times more than the image candidates in image retrieval; and (iii) The similarity computation of SCAN relies on the cross-attention computation, which is straightforward to be paralleled, even in their official code; while our FILIP's similarity computation is simply a matrix multiplication and is readily optimized for most modern hardware.
>
> Please also refer to Q4 of reviewer 2 for more discussions about the inference time of different methods.

---

> > ### Author Response · Authors · 2021-11-21
> > **Reply to Reviewer 3**
> >
> > **Q3: The large-scale V-L pre-training dataset are not public available. We cannot sure whether there are similar images/texts between the large-scale pre-training dataset (training + validation split) and the f30k/coco baseline dataset (val/test split). If the answer is true, then it is definitely unfair.**
> >
> > **Reply:** It is time-consuming and expensive to collect and clean such a multi-modality dataset at our scale. However, for fair comparison, we plan to release a dataset with hundreds of millions image-text pairs and the inference code if the paper is accepted, for purpose of improvement of this community. Furthermore, as we understand the large-scale pre-training may be too costly for colleges and individual researchers, we plan to release some pre-trained models for benchmarking the performance of large-scale multi-modal models as well as facilitating more research on downstream tasks and applications.
> >
> > Besides, the comparison in Table 2 is fair since we follow a similar method to collect data as the ALIGN and CLIP (current two SOTA zero-shot Image-text retrieval methods). Note that when collecting the images and text from the Internet, no human annotation is needed anymore.
> >
> > **Q4: The SoTA performance are mainly come from 2 aspects: (i) The FILIP300M large-scale pre-training dataset. (ii) Grid/Patch feature (1 image, multiple features) methods like ViTs, which are the latest advances of backbone networks in CV. So, it’s SoTA performance should not be largely due to the proposed cross-modal late interaction.**
> >
> > **Reply:** (1) The size of our pre-training dataset is smaller comparing to CLIP and ALIGN (340M VS 400M/1800M). However, with the proposed cross-modal late interaction, our models can outperform them in most down-steam tasks (with the same model architectures as CLIP).
> >
> > (2) Note that CLIP also uses ViT as image encoders. Our model architecture of FILIP is exactly the same as CLIP as mentioned in Section 4.1, except we use a different loss. So our performance gain over CLIP is not due to the use of ViTs.
> >
> > (3) To verify that the performance gain comes from the proposed cross-modal late interaction, we refer the reviewer to the ablation study on YFCC in Table 4. With the same pre-training data and the exactly same image encoder as CLIP with Grid/Patch feature,  replacing the original loss in CLIP (i.e., Baseline (ViT-B/32)) to the proposed cross-modal late interaction loss significantly improves the performance, with an absolute R@1 gain of 5.5\% (resp. 3.8\%) for image-to-text (resp. text-to-image) retrieval on MSCOCO and an absolute top-1 accuracy gain of 3.9\% for zero-shot classification on ImageNet.
> >
> > (4) To further verify our statements, we train CLIP's ViT-B-32 model (referred as $\rm CLIP_{rep}
> > $) using the same training dataset as our FILIP and evaluate it using the same prompts on 12 downstream classification datasets. The results are shown in the following table (also updated in Appendix A.7).  With the same architecture, the $\rm FILIP_{base}$ has 6.7 points higher average accuracy than the  $\rm CLIP_{rep}$ over 12 datasets. This further verifies that the performance gain come from the proposed cross-modal late interaction, rather than the data or architecture.
> >
> > | model              | CIFAR10 | CIFAR100 | Caltech101 | StanfordCar | Flowers102 | Food101 | SUN397 | DTD  | Aircrafts |
> > | ------------------ | ------- | -------- | ---------- | ----------- | ---------- | ------- | ------ | ---- | --------- |
> > | CLIP               | 91.3    | 65.1     | 87.9       | 59.4        | 66.7       | 84.4    | 63.2   | 44.5 | 21.2      |
> > | $\rm CLIP_{rep}$   | 82.0    | 57.5     | 89.9       | 45.1        | 80.7       | 75.1    | 63.6   | 46.7 | 33.7      |
> > | $\rm FILIP_{base}$ | 86.9    | 65.5     | 91.9       | 55.4        | 85.3       | 82.8    | 69.1   | 49.3 | 57.2      |
> >
> > | model              | OxfordPet | EuroSAT | ILSVRC | Average  |
> > | ------------------ | --------- | ------- | ------ | -------- |
> > | CLIP               | 87        | 49.4    | 63.2   | 64.9     |
> > | $\rm CLIP_{rep}$   | 82.7      | 49.0    | 64.2   | 63.8     |
> > | $\rm FILIP_{base}$ | 88.1      | 49.9    | 68.8   | **71.5** |
> >
> > **Q5:  The shortcoming of ViTs may be Inherited by FILIP. So, the training/testing speed may not exceed CLIP/ALIGN. Can the authors give more explanations?**
> >
> > **Reply:** (1) CLIP also uses ViT as image encoders. Our model architecture of FILIP is exactly the same as CLIP as mentioned in Section 4.1, except that we use a different loss and a different training strategy. And our models have similar inference speed as CLIP, see Appendix A.8.
> >
> > (2) Besides ViTs, CLIP also uses ResNets as an option for image encoders, but find them less efficient than ViTs as summarized in their Section 3.2 based on their Figure 10: "We also find that CLIP vision transformers are about 3x more compute efficient than CLIP ResNets, which allows us to reach higher overall performance within our compute budget."

---

> > > ### Author Response · Authors · 2021-11-21
> > > **Reply to Reviewer 3**
> > >
> > > **Q6: The statements in this paper have a double standard on "offline/pre-computing" . (i) For other works: “This design complicates ... due to pre-computing and storing a large number of ROI features” (page 1). (ii) For their work: "FILIP successfully ... while simultaneously ... to pre-compute image and text representations offline" (page 1) Therefore, we don’t know weather offline/pre-computing is efficient or not?**
> > >
> > > **Reply:** (1) Note that the two offline precompute are performed for different purposes (i.e., one for pre-training and the other for inference) and have totally different roles.  Specifically, (i) For the object detector mentioned in page 1, offline precomputing ROI features (about 200 ROI features per image) complicates **pre-training** due to the infeasible storage and huge amount of precomputing of these features (please refer to our reply to Q3 for Reviewer 2)
> > > (ii) For the large-scale retrieval of dual-stream models, offline precomputing features greatly speeds up the retrieval task at **inference** time compared to single-stream models ($2N$ vs. $N^2$)  (Please refer to our reply to Q4 for Reviewer 2).
> > >
> > > **Q7: There are some inappropriate expressions/mistakes in the introduction (page 1). (i) “One line of work” can be substitute by “Bottom-up attention [2] (see below) methods”. (ii) “Another line of work” can be replaced by “Cross-attention methods”. (iii) “self-attention (Kim et al., 2021)” is inappropriate, because ViLT is a typical cross-attention method (see Figure 3, the cross-modal Transformer Encoder), rather than (pure) self-attention.**
> > >
> > > **Reply:** The reviewer may have some misunderstanding on "self-attention" and "cross-attention" under the context of Transformer-models,  which are the defacto backbones of most VLP models discussed in the introduction/related work, and also our proposed FILIP.
> > >
> > > (1) Under this context, if the image region/patch features are **concatenated** as a single sequence before being fed into the Transformer, then they use **self-attention** for cross-modal interaction. On the other hand, if the image features are interacted with textual tokens through the encoder-decoder attention in the decoder of an encoder-decoder Transformer, then they use **cross-attention** (please refer to ALBEF(Li et al., 2021)) for cross-modal interaction.
> > >
> > > (2) Indeed, ViLT (Kim et al., 2021) does not mention "cross-attention" at all in their paper, but mentions "self-attention" in their "Model Overview"part in Section 3.1. On the other hand, "cross-attention" is used in ALBEF (Li et al., 2021) to denote the encoder-decoder attention in the computation of the decoder (Please refer to their model architecture in Figure 1).
> > >
> > > **References:**
> > >
> > > [1] Kuang-Huei Lee, Xi Chen, Gang Hua, Houdong Hu, and Xiaodong He. Stacked cross attention for image-text matching. In ECCV, 2018.
> > >
> > > [2] Kim, Wonjae, et al. "Vilt: Vision-and-language transformer without convolution or region supervision." In ICML, 2021.
> > >
> > > [3] Li, Junnan, et al. "Align before fuse: Vision and language representation learning with momentum distillation." In NeurIPS, 2021.

---

> > > > ### Author Response · Authors · 2021-12-09
> > > > **Reply to Reviewer 3**
> > > >
> > > > Thanks for the detailed comments. We have addressed your further concerns and clarified your points in the reply above. Do you have an updated assessment of our paper? Thanks for your consideration.

---

> > ### Comment · Reviewer_gJkL · 2021-11-30
> > **Reply to Authors**
> >
> > Thanks for the detailed responses. However, replacing the bottom-up features with patch representation via ViT seems not new, what about replacing the faster RCNN with ViT in SCAN?  About the efficiency comparision, did the authors also use FP16 features and dimensionality reduction techiques to the compared CLIP and SCAN to make a fair comparison?

---

> > > ### Author Response · Authors · 2021-11-30
> > > **Reply to Reviewer 3**
> > >
> > > **Q8: However, replacing the bottom-up features with patch representation via ViT seems not new, what about replacing the faster RCNN with ViT in SCAN?**
> > >
> > > **Reply:** Our key contribution of this work does not lie in the dual-stream model architecture (including the use of ViT) itself, but lies in making cross-modal late interaction tractable in **large-scale contrastive learning** for vision-language pre-training, to boost the downstream tasks' performance and simultaneously allows **fine-grained word-patch alignment**, which is not explored in previous works.  Besides, replacing the faster RCNN with ViT in SCAN is non-trivial. The pretrained ViT model from ImageNet does not have meaningful patch representations because the classification loss is only built on the classification token in ViT. Thus, directly replacing faster RCNN with pretrained ViT may not lead to any localization ability of fine-grained ability.
> > >
> > >
> > >
> > > **Q9: About the efficiency comparison, did the authors also use FP16 features and dimensionality reduction techniques to the compared CLIP and SCAN to make a fair comparison?**
> > >
> > > **Reply:** As is suggested by the reviewer, we also test the inference time of CLIP (ViT-L/14) and SCAN with FP16 features and the same feature dimension 256 as FILIP. The other settings are the same as that in Appendix A.8. The results are shown below.
> > >
> > > | model | i-to-t (Flickr30K) | t-to-i (Flickr30K) | i-to-t (MSCOCO) | t-to-i (MSCOCO) |
> > > | ----- | ------------------ | ------------------ | --------------- | --------------- |
> > > | SCAN  | 3977 ms            | 4 ms               | 20362 ms        | 6 ms            |
> > > | CLIP  | 23 ms              | 8 ms               | 18 ms           | 9 ms            |
> > > | FILIP | 24 ms              | 8 ms               | 26 ms           | 9 ms            |
> > >
> > > For CLIP, we use FP16 global feature for similarity calculation and reduce the feature dimension from 768 to 256 as FILIP. As can be seen, these two modifications  have negligible influence on CLIP. This is because computing only the global similarity in CLIP is much cheaper than running the model. Specifically, consider image-to-text retrieval on MSCOCO dataset with 25,000 candidate texts, which has the most (i.e., 25000) candidates and is most expensive in similarity calculation among the four retrieval tasks studied. We use Floating Point Operations (FLOPs) to measure the computation complexity. (i) For computing the global similarity between one image query and 25000 candidate texts, the FLOPs required is only $2\times768\times25000=38.4M$ even for FP32 features.  (ii) On the other hand, the FLOPs required for running an image with sequence length $n=256$ through the 24-layer (i.e., $L=24$) image encoder VIT-L/14 with hidden dimension $d=1024$ is over $2L(2n^2d+4nd^2+2nd \times 4d)=161G$ for only the matrix multiplications involved, which is over 3 magnitudes more expensive than computing the similarity.
> > >
> > > For SCAN, we run the model with the official released code. We use FP16 feature for cross-attention computation, and reduce the feature dimension from 1024 to 256. By comparison with Table 17 in Appendix A.8, these two techniques improves the inference speed of SCAN on all four tasks. Benefiting from the light GRU text encoder, SCAN is slightly faster than FILIP on text-to-image retrieval tasks. Please also note that  the original SCAN with FP32 features and a larger dimension (i.e., 1024) already has much worse retrieval performance than FILIP (i.e. for text-to-image retrieval, 48.6 R@1 VS 87.1 on Flickr30K,  38.6 R@1 VS 61.2  on MSCOCO), using these two techniques endures a risk of enlarging the performance gap between SCAN and FILIP. Moreover,  SCAN is still much slower than FILIP on image-to-text retrieval on Flickr30K, similar to the observation in Table 17.

---

### Official Review · Reviewer_WLou · 2021-11-02

**Correctness:** 3
**Technical Novelty And Significance:** 2
**Empirical Novelty And Significance:** 3
**Recommendation:** 6
**Confidence:** 4

**Main Review:**

Strengths:

1. The paper is quite well written and it is easy to follow the authors.
2. The approach is simple yet efficient. It seems natural to extend the prior clip and align the approach with fine-grained interactions.
3) New large-scale image-text pair dataset called FILIP300M would be valuable to the community.
4) Experiments show strong performance on vision-language tasks.

Weaknesses/Concerns:

1) Model novelty is limited considering (I) similarity to  Khattab & Zaharia, and (ii) prior Vision-Language Pre-training models.

2) On a similar note to the previous point, the paper states that they follow Khattab & Zaharia (2020). The changes to the proposed compared to Khattab & Zaharia seem trivial and nothing especially novel except applying to a new task. I would request the authors to discuss this in further detail to make the paper contribution clearer. The only thing the paper states 'we discard the padded tokens and use average instead summation of token-wise maximum similarities when computing the image-text alignment, which enhances the cross-modal representation learning and stabilizes training. Furthermore, we construct a large-scale pre-training dataset named FILIP300M from the Internet.'  -- This seems a minimal change. Also, where are the experimental results comparing the paper and supporting these arguments - enhanced learning and stabilized training?

3) Ablation: The paper states that the use of object detector-based approaches leads to scalability issues in training vision-language models. I do not expect that to create a significant issue based on prior experience. Did other papers report also the same? Or, Did the authors' experiment and find the issue? How much impact it has?
On a related note, it is also important to see - How the performance changes when a faster rcnn based detector is used to understand the tradeoff?

4) As the paper is interested in efficient inference, it is important that the paper include inference time for compared methods, especially in Table 2 and Table 3.

5) An approach to get fine-grained interaction between visual and textual modalities avoiding the problem of pre-computing features offline), is not novel in general. As far as I am aware, such ideas are especially not uncommon in video-text retrieval. For example, prior work(Weakly Supervised Video Moment Retrieval From Text Queries, CVPR 2019) adopts a late visual-language interaction strategy with weighted pooling for video-text retrieval. However, the proposed fine-grained interaction approach seems to be new with recent transformer-based vision-language models.

6) MInor: Did the authors consider - For a visual token, similarity with the full-text representation for fine-grained interaction (as a visual patch may be related to full text or multiple words, not only one word token)? Compared to For a visual token, similarity with all textual tokens (i.e., what used in the paper). Any comment on this?

7) Will the dataset be publicly released?

**Summary Of The Paper:**

The paper proposes an approach for large-scale vision-language representation learning that focuses on achieving fine alignment between modalities through a late interaction strategy.  Experiments on zero-shot image classification and image-text retrieval tasks show that the proposed approach is able to achieve state-of-the-art performance. A large-scale image-text dataset is also constructed from the web for pre-training.

**Summary Of The Review:**

The paper is well-written with strong results. However, the proposed approach has limited novelty and some critical claims are made without strong support.

---

> ### Author Response · Authors · 2021-11-21
> **Reply to Reviewer 2**
>
> **Q1: Model novelty is limited considering (i) similarity to Khattab \& Zaharia, and (ii) prior Vision-Language Pre-training models.**
>
> **Reply:** (1) Please refer to our reply to Q1 of reviewer 1 for the novelty of this work and differences with Khattab \& Zaharia, (2020).
>
> (2) The main difference with prior large-scale dual-stream Vision-Language Pre-training Models lies in the usage of cross-modal late interaction which learns finer-granular word-patch alignment and achieves SOTA performance on various tasks. From Figure 2, it can be found that our method can localize the objects in the image and automatically learn word-patch alignment.
>
> **Q2: On a similar note to the previous point, the paper states that they follow Khattab \& Zaharia (2020). The changes to the proposed compared to Khattab \& Zaharia seem trivial and nothing especially novel except applying to a new task. I would request the authors to discuss this in further detail to make the paper contribution clearer. The only thing the paper states 'we discard the padded tokens and use average instead summation of token-wise maximum similarities when computing the image-text alignment, which enhances the cross-modal representation learning and stabilizes training. Furthermore, we construct a large-scale pre-training dataset named FILIP300M from the Internet.' -- This seems a minimal change. Also, where are the experimental results comparing the paper and supporting these arguments - enhanced learning and stabilized training?**
>
> **Reply:** Thanks for the suggestion. We add both a slight discussion in Section 3.1.1 on page 5, and detailed supporting experiments in Appendix A.6, to show the effect of the two changes (i.e., remove padded tokens and use average instead of sum). The experimental results support the arguments "enhanced learning and stabilized training" by showing that  these two changes are crucial to the good performance on downstream tasks and the quality of the learned finer-granular word-patch alignment.
>
> **Q3: Ablation: The paper states that the use of object detector-based approaches leads to scalability issues in training vision-language models. I do not expect that to create a significant issue based on prior experience. Did other papers report also the same? Or, Did the authors' experiment and find the issue? How much impact it has? On a related note, it is also important to see - How the performance changes when a faster rcnn based detector is used to understand the tradeoff?**
>
> **Reply:** (1) Scalability issues for detector-based approaches: a) Online inference of the detector is not tractable for large-scale training, i.e., a FasterRCNN detector processes about 20 FPS per GPU card, which greatly slows down the training and requires more GPU memory.
> Thus, to improve  the efficiency of large-scale pre-training, a common method is to precompute and store the features for each bounding box proposal for all the pretraining data offline and then pre-train with these features. However, storing these features of 340M images require 8KB x 200 (bbox per img) x 340M (imgs) = 550TB hard disk space, which is memory expensive.
>
> (2) The detector is usually pre-trained on a predefined number of categories (e.g. about 80 for MSCOCO and 3000 for VG). However, for large-scale pretraining, those categories may not be enough. Besides, the performance of those rare categories is bad and cannot meet the needs of zero-shot tasks. The very recent paper simVLM from Google has a similar opinion for this issue: "Creating a bottleneck for further quality improvement. What is more, such pretraining-finetuning based approaches usually lack the zero-shot capability".
>
> (3) For the comparison of performance, from Table 3, our method also outperforms some detector-based approaches such as VinVL and UNIMO (78.9 R@1 VS 75.4 \& 65.7 on MSCOCO).

---

> > ### Author Response · Authors · 2021-11-21
> > **Reply to Reviewer 2**
> >
> > **Q4: As the paper is interested in efficient inference, it is important that the paper include inference time for compared methods, especially in Table 2 and Table 3.**
> >
> > **Reply:** (1) Theoretically:
> >
> > Consider a simple text-to-image retrieval system with an index consisting of N images. For single-stream models like UNITER and OSCAR, for each text query, the inference computation includes running the model N times, each time with a concatenated sequence of the text query and an image. On the other hand, for dual-stream models like CLIP, Align and FILIP, one first stores the features of these images computed using the image encoder for only once offline. Then for each text query, the inference computation consists of only i) running the text encoder once for the text query and ii) N similarity computation between the text feature and pre-computed image features. Note that both (i) and (ii) are much cheaper and are negligible compared to running the model N times (Refer to our reply to Q7 of Reviewer 4). This is also why the inference with dual-stream models is much cheaper than single-stream model.
> > The inference efficiency comparison between dual-stream and single-stream models is also widely studied in the literature, in the domain of information retrieval (Khattab\&Zaharia,2020), as well as image-text retrieval in ALIGN (Jia et al., 2021).
> >
> > (2) Empirically:
> > ALIGN (Jia et al., 2021) empirically show in a real retrieval system that the single-stream model UNITER and OSCAR are several magnitudes slower than dual-stream models: "Recently more advanced models emerge with cross-modal attention layers (Lu et al.,2019; Chen et al., 2020c) and show superior performance in image-text matching tasks. However, they are orders of magnitudes slower and hence impractical for image-text retrieval systems in the real world."
> >
> > In Appendix A.8, we compare the inference time of our proposed method with CLIP on image-text retrieval. Compared to CLIP, our proposed FILIP is only slightly slower for image-to-text retrieval in the inference due to token-wise similarity computation, but should still be much faster than single-stream models which are several magnitudes slower (Jia et al., 2021).
> >
> > **Q5: An approach to get fine-grained interaction between visual and textual modalities avoiding the problem of pre-computing features offline), is not novel in general. As far as I am aware, such ideas are especially not uncommon in video-text retrieval. For example, prior work(Weakly Supervised Video Moment Retrieval From Text Queries, CVPR 2019) adopts a late visual-language interaction strategy with weighted pooling for video-text retrieval. However, the proposed fine-grained interaction approach seems to be new with recent transformer-based vision-language models.**
> >
> > **Reply:** Our key contribution of this work does not lie in the dual-stream model architecture itself, but lies in making  cross-modal late interaction tractable in contrastive learning for vision-language pre-training, to boost the downstream tasks' performance and simultaneously allows fine-grained word-patch alignment during contrastive learning, which is not explored in previous works. Furthermore, this is the first work to explore this line of design in a 100M+ image-text scale, which has been proved useful. Please also refer to our reply to the first question of Reviewer 1, where we discussed in detail about the novelty of this work.
> >
> > **Q6: Minor: Did the authors consider - For a visual token, similarity with the full-text representation for fine-grained interaction (as a visual patch may be related to full text or multiple words, not only one word token)? Compared to For a visual token, similarity with all textual tokens (i.e., what used in the paper). Any comment on this?**
> >
> > **Reply:** (1) We agree with the reviewer that a visual patch may be related to more than one textual token. From the visualizations in Section 4.4, even for one object "the white butterfly", its class name is tokenized to more than one token, and the image patches of this object are can be classified to either "white" or "butterfly".
> > (2) Similar to the reviewer's suggestion, we are exploring a multi-scale extension of current word-patch alignment to  capture alignment between image patches with more global textual information. We include it as a future work.
> >
> > **Q7: Will the dataset be publicly released?**
> >
> > **Reply:** It is time-consuming and expensive to collect and clean such a multi-modality dataset at our scale. However, we plan to release a dataset with hundreds of millions image-text pairs if the paper is accepted, for purpose fair comparison and improvement of this community. Furthermore, as we understand the large-scale pre-training may be too costly for colleges and individual researchers, we plan to release some pre-trained models for benchmarking the performance of large-scale multi-modal models as well as facilitating more research on downstream tasks and applications.

---

> > > ### Comment · Reviewer_WLou · 2021-11-29
> > > **Response to Rebuttal**
> > >
> > > I would like to thank the authors for their response and the improvements/additions made to the paper. They have addressed most of my concerns. It is also good to know that the authors plan to release the dataset.

---

### Official Review · Reviewer_aFb7 · 2021-11-03

**Correctness:** 4
**Technical Novelty And Significance:** 3
**Empirical Novelty And Significance:** 4
**Recommendation:** 6
**Confidence:** 4

**Main Review:**

Strengths:
1. Strong performance on zero-shot classification and image-text retrieval, compared with original CLIP. A fair comparison with CLIP pre-trained on the YFCC dataset.
2. The investigation on token-level interaction to provide a more fine-grained contrastive signal in Language-Image Pre-training is interesting and meaningful for future research in this direction.

Weakness:
1. Novelty in total is limited. The proposed cross-modal late interaction is inspired by [1], but the three modifications mentioned in the paper seem to be a bit trivial. And image and text augmentation are widely used in previous works.
2. The study on token-level interaction is not comprehensive in terms of methodology. The paper only uses max-mean late interaction without exploring different variants. For example, how will the mean-mean, max-max, or mean-max interaction work? One step further, how about we ensemble the token features inside each modality with a set of learnable weights and then compute the similarity between ensembled features?
3. One type of experiment is missing - linear probing on the visual encoder. This is to prove the transferable representation learned by Language-Image Pre-training.
4. The authors do not mention whether to release their collected dataset FILIP300M. If not, then it is hard to fairly compete for researchers interested in this direction.


Ref:
[1] Omar Khattab and Matei Zaharia. Colbert: Efficient and effective passage search via contextualized late interaction over bert. In International ACMSIGIR Conference on Research and Development in Information Retrieval, pp. 39–48, 2020

**Summary Of The Paper:**

The paper proposes to utilize the fine-grained alignment between visual tokens and text tokens in the contrastive loss for language-image pertaining. More specifically, the similarities between images and captions are calculated by averaging the token-wise maximum similarities. The experiments on both zero-shot image classification and image-text retrieval with different pre-trained datasets validate the effectiveness of the proposed model.

**Summary Of The Review:**

The main drawback of this paper is that the proposed model is not that novel in terms of methodology. But this paper provides an angle to look at the token-wise interaction in contrastive Language-Image pre-training. Moreover, the performance is quite impressive and the experiments are conducted on different pre-training datasets and downstream tasks to prove the model's efficacy. Overall, I think it is marginally above the threshold. I may change my mind based on the authors' rebuttal and other reviewers' feedback.

---

> ### Author Response · Authors · 2021-11-21
> **Reply to Reviewer 1**
>
> **Q1: Novelty in total is limited. The proposed cross-modal late interaction is inspired by [1], but the three modifications mentioned in the paper seem to be a bit trivial.**
>
> **Reply:** (1) Recent dual-stream models (e.g., CLIP and ALIGN)  lack the ability to capture the relationship between visual objects and textual words, thus we propose the cross-modal late interaction mechanism to solve this problem, which is the first trial in vision-language multi-modal learning. Besides, we empirically show that unsupervised training with this simple late interaction works surprisingly well in word-patch alignment and learns meaningful fine-grained features with promising localization ability, as illustrated in Section 4.5. It is worthy to note that this kind of token-level alignment is not found in [1].
>
> (2) As far as we know, it is the first time such token-level late interaction is used in large-scale contrastive learning. This is not trivial as a direct  application will cause severe efficiency problems, and make large-scale pre-training intractable. Specifically, the late interaction requires computing token-wise similarities which can be inefficient in terms of communication, memory and computation, especially when the number of negatives (\# batch size-1) is large. To make it work, we made quite several attempts and optimizations as stated in "the Training Efficiency" paragraph at the end of Section 3.1.1 and verified empirically in "Efficiency Study of Cross-modal Late Interaction" in Section 4.4.
>
> (3) The other two modifications (i.e., remove padded tokens and use average instead of sum) though seem simple, but are also crucial to the performance. When these two modifications are removed, for $\rm FILIP_{base}$ trained on filtered YFCC100M, the zero-shot Top1 accuracy of ImageNet drops from 34.3 to 32.7. More experimental settings can be found in Appendix A.6. Please also refer to our reply to Q2 for ablation on other token-level variants, which also show that these two modifications are important for the final performance.
>
> (4) The two modifications are also crucial to the quality of finer-granular word-patch alignment. Besides the performance drop, according to the visualizations in Appendix A.6, using the original late interaction in [1] leads to less accurate word-patch alignment.
> The object patches are often aligned to the padded tokens instead of class names as the padded tokens learn similar representations as as existing key textual tokens, similar to the finding in Section 3.2 of [1] that padding with masked tokens (which is called "query augmentation" in [1]) tend to "re-weigh existing terms based on their importance for matching the query".
>
> **Q2: The study on token-level interaction is not comprehensive in terms of methodology. The paper only uses max-mean late interaction without exploring different variants. For example, how will the mean-mean, max-max, or mean-max interaction work? One step further, how about we ensemble the token features inside each modality with a set of learnable weights and then compute the similarity between ensembled features?**
>
> **Reply:** Thanks for the suggestion, we add a comparison with the suggested variants in the following table. We replace the cross-modal late interaction of $\rm FILIP_{base}$ with these variants. We train on filtered YFCC100M using mixed-precision training on 8 V100 GPUs, with a batch size of 512 per GPU and embedding dim 256. For "weighted mean", we ensemble the token-wise features with a set of learnable weights (i.e., 77 weights for 77 textual tokens and 49 weights for 49 image patches) and then use the weighted average representation as the global feature of each modality to compute token-wise similarities. As can be seen, the other variants perform significantly worse than the "max-mean" used in our proposed method. Besides the worse performance, these variants also can not learn meaningful word-patch alignment.
>
> |                      | ours(max-mean) | mean-mean | max-max | mean-max | weighted mean |
> | :------------------: | :------------: | :-------: | :-----: | :------: | :-----------: |
> | ImageNet Top1 Acc(%) |      **35.2**      |   16.2    |  30.8   |   30.4   |     31.1      |

---

> > ### Author Response · Authors · 2021-11-21
> > **Reply to Reviewer 1**
> >
> > **Q3:  One type of experiment is missing - linear probing on the visual encoder.  This is to prove the transferable representation learned by Language-Image Pre-training.**
> >
> > **Reply:** Thanks for the suggestion. We add the linear probing experiments on 12 datasets in Appendix A.5. The following table shows the results. FILIP outperforms CLIP with an average improvement of 2% in base model and 1.5% in large model.
> >
> > | model               | CIFAR10 | CIFAR100 | Caltech101 | StanfordCar | Flowers102 | Food101 | SUN397 | DTD  | Aircrafts |
> > | ------------------- | ------- | -------- | ---------- | ----------- | ---------- | ------- | ------ | ---- | --------- |
> > | CLIP-ViT-B/32       | 95.1    | 80.5     | 93         | 81.8        | 96.6       | 88.8    | 76.6   | 76.5 | 52        |
> > | $\rm FILIP_{base}$  | 95.3    | 80.3     | 95         | 78.6        | 98.7       | 86.2    | 77.9   | 78.1 | 76.6      |
> > | CLIP-ViT-L/14       | 98      | 87.5     | 96.5       | 90.9        | 99.2       | 95.2    | 81.8   | 82.1 | 69.4      |
> > | $\rm FILIP_{large}$ | 97.9    | 87       | 97.2       | 89          | 99.6       | 94.6    | 83.2   | 83.9 | 84.8      |
> >
> > | model               | OxfordPet | EuroSAT | ILSVRC | Average  |
> > | ------------------- | --------- | ------- | ------ | -------- |
> > | CLIP-ViT-B/32       | 90        | 97      | 76.1   | 83.7     |
> > | $\rm FILIP_{base}$  | 88        | 97.5    | 75.8   | **85.7** |
> > | CLIP-ViT-L/14       | 95.1      | 98.2    | 83.9   | 89.8     |
> > | $\rm FILIP_{large}$ | 93.5      | 100     | 84.5   | **91.3** |
> >
> > **Q4: The authors do not mention whether to release their collected dataset FILIP300M. If not, then it is hard to fairly compete for researchers interested in this direction.**
> >
> > **Reply:** Thanks for the suggestion. It is time-consuming and expensive to collect and clean such a multi-modal dataset at our scale.
> > However, we plan to release a dataset with hundreds of millions image-text pairs and the inference code if the paper is accepted,
> > for purpose of fair comparison with other methods and improvement of the community. Furthermore, as we understand the large-scale pre-training may be too costly for colleges and individual researchers, we plan to release some pre-trained models for benchmarking the performance of large-scale multi-modal models as well as  facilitating more research on downstream tasks and applications.
> >
> > **Reference:**
> >
> >  [1] Omar Khattab and Matei Zaharia. Colbert: Efficient and effective passage search via contextualized late interaction over bert. In International ACMSIGIR Conference on Research and Development in Information Retrieval, pp. 39–48, 2020

---

> > > ### Comment · Reviewer_aFb7 · 2021-11-30
> > > **Thank authors for the response.**
> > >
> > > Thank you for the rebuttal. The added experiments are comprehensive and solve most of my concerns. It is also great to hear that the authors plan to release the dataset and provide the benchmarks.

---

### Author Response · Authors · 2021-11-21
**General Response to All reviewers**

We thank all the reviewers for their insightful and valuable comments. Firstly, we would like to summarize the novelty of this work (detailed supporting experiments/discussions can be found in our reply to Reviewer 1) which is questioned by all reviewers.

1. Recent dual-stream VLP models lack fine-grained alignment between visual objects and words, thus we propose to
   apply a cross-modal late interaction mechanism to solve this problem, which is the first trial in vision-language multi-modal pre-training. We also empirically show that our unsupervised training works surprisingly well in word-patch alignment and learns meaningful fine-grained features with promising localization ability.
2. This is the first time such kind of token-level late interaction is used in large-scale contrastive learning.
   This is not trivial as a direct use will cause severe efficiency problems, and we make quite abundant attempts and optimizations on cluster servers (e.g., use FP16 features, reduce feature dimension, and select top 25% tokens) to make it work.
3. The other two modifications (i.e., remove padded tokens and use average instead of sum) are crucial to the good performance on downstream tasks (e.g., FILIP outperforms the well-known CLIP model by 3 - 5% in terms of the average image classification accuracy on 12 tasks.) and the quality of the learned finer-granular word-patch alignment, which sheds lights on future efficient dual-stream VLP in this direction.
4. We also contribute a new large-scale multi-modal dataset and will release the pretrained models for benchmarking the performance of large-scale multi-modal pretraining and facilitating more downstream tasks and applications.

Then in our reply to each reviewer, we address each reviewer's detailed  concerns point by point. We have revised the manuscript according to reviewers' comments. The main changes (highlighted in blue) we made include:

1. In Appendix A.5, we add an experiment of linear probing on the visual encoder of FILIP;
2. In Appendix A.6, We add an experiment comparing the proposed cross-modal late interaction and the original one in KHATTAB \& ZAHARIA (2020), in terms of downstream tasks' performance and the word-patch alignment.
3. In Appendix A.7, we add an experiment comparing the proposed cross-modal late interaction with CLIP's loss on the full pre-training dataset.
4. In Appendix A.8, we report the inference time of different methods on image-text retrieval.
5. In related work, we added some more discussion about the connection and difference between the proposed method and one reference mentioned by Reviewer 3.

---

### Decision · Program_Chairs · 2022-01-20

**Decision:**

Accept (Poster)

**Comment:**

The focus on this paper's proposed FILIP method is to perform word-patch alignment by token-wise similarity matrix through cross-modal late interaction by modifying only contrastive loss, leading to training and inference efficiency. The authors also collected FILIP300M, a large-scale cleaned image captioning dataset for FILIP’s V-L pre-training. FILIP achieves strong performance on zero-shot image classification and image-text-matching tasks, and the paper also visualizes the ability of fine-grained (visual-textual token) classification and localization. Overall most of the reviewers appreciated the idea and the generalization results, but had some concerns about not enough technical novelty over the Khattab and Zaharia Colbert paper, which this paper adopts for multimodal tasks. Some reviewers also had concerns about the dataset release but the authors promise to address this. Some reviewers were also not fully convinced about the high storage requirements and scalability for some of the retrieval tasks that the authors tested.

NOTE: The authors are also asked to describe any ethical considerations or issues that arise in their large scale dataset collection in the camera ready version of the paper, see https://arxiv.org/abs/2110.01963 for examples.